# Exploring the Potential Processes Controls for Changes of Precipitation-Runoff Relationships in Non-stationary Environments

Tian Lan[1,2,3], Tongfang Li[1,2,3*], Hongbo Zhang[1,2,3*], Jiefeng Wu[4], Yongqin David Chen[5], Chong-Yu Xu[6]

[1]School of Water and Environment, Chang'an University, Xi'an 710054, China.

5 [2]Key Laboratory of Subsurface Hydrology and Ecological Effects in Arid Region of the Ministry of Education, Chang'an University, Xi'an 710054, China.

[3]Key Laboratory of Eco-Hydrology and Water Security in Arid and Semi-Arid Regions of Ministry of Water Resources, Chang'an University, Xi'an 710054, China.

[4]School of Hydrology and Water Resources, Nanjing University of Information Science and Technology, Nanjing 210000, Jiangsu,

10 China.

[5]School of Humanities and Social Science, The Chinese University of Hong Kong, Shenzhen 518172, China.

[6]Department of Geosciences, University of Oslo, P.O. Box 1047 Blindern, 0316 Oslo, Norway.

*Correspondence to*: Tongfang Li (tongfangli@chd.edu.cn), Hongbo Zhang (hbzhang@chd.edu.cn)

## Abstract

The influence of climate change and anthropogenic activities on precipitation-runoff relationships (PRR) has been widely discussed. Traditional models assuming stationary conditions can lead to inaccurate streamflow predictions. To address this issue, we propose a Driving index for changes in Precipitation-Runoff Relationships (DPRR), identified as key PRR influencers, involving climate forcing, groundwater, vegetation dynamics, and anthropogenic influences. According to the quantitative results of inputting the candidate driving factors to a holistic conceptual model, the possible process explanations for changes in the PRR were deduced. This framework is validated across five sub-basins in the Wei River Basin. Moreover, non-stationary hydrological processes were initially detected, and the nonlinear correlations among the factors were assessed. The results show that baseflow emerges as the primary factor positively influencing PRR (enhancing PRR), but with high uncertainty. Potential evapotranspiration plays a dominant role in driving negative PRR changes in the sub-basins which are characterized by a semi-arid climate and minor human interference. Vegetation dynamics negatively influence PRR, with driving levels correlating with the scale of soil and water conservation engineering, displaying lower uncertainty. Anthropogenic influences, represented by Impervious Surface Ratio (ISR), Night-Time Light (NTL), and population density (POP), exhibit varying driving levels, with ISR having the strongest and direct impact, closely linked to urbanization processes and scale. The temporal dynamics of driving factors computed by dynamic DPRR generally correspond with hydrological regime shifts in non-stationary environments. The study's findings offer a comprehensive understanding of hydrological processes, enabling informed decision-making for the development of sustainable hydrological models.

## 1 Introduction

Precipitation-Runoff Relationships (PRR) are a crucial concern within the fields of engineering hydrology, water resource planning and management, and catchment system evolution (Saft et al., 2015; Nourani et al., 2016; Zhang et al., 2018). As foundational elements of the hydrological cycle, precipitation and runoff are interconnected, with runoff acting as a temporal/spatial response to precipitation (Xu et al., 2010). This linkage is influenced by climatic and land-cover attributes, such as catchment size, topography, soil composition, vegetation, and spatial heterogeneity (Bales et al., 2018). Traditionally, these attributes have been considered stationary in catchments with minimal human disturbances (Tian et al., 2018). However, climate change, dam construction, vegetative restoration, urbanization, and increased water consumption have induced complexities and non-stationarities in many regional streamflow patterns (Dey and Mishra, 2017). As a result, the relationships between precipitation and runoff can exhibit complexity and non-stationarity (Franzen et al., 2020). However, prevailing modeling approaches often maintain assumptions of stationary climatic and catchment attributes (Pathiraja et al., 2016), which limits their applicability in non-stationary environments. For example, in long-term runoff simulations, non-stationary climate conditions and catchment characteristics may lead to serious discrepancy between simulated runoff based on fixed parameters and observed runoff, thus affecting water resources planning and operations. This suggests a need for identifying the potential driving mechanisms behind these changes or non-stationary hydrological conditions. These efforts can contribute to enhancing predictive modeling, reservoir management, water resources decision-making, and the preservation of ecosystem function and services (Hejazi and Cai, 2009; Quinn et al., 2017).

Various methods have been employed to describe the PRR, demonstrating successful applications. The correlation coefficient method is the commonly employed technique to identify the PRR due to its simplicity and effectiveness. However, certain limitations of correlation coefficients are often overlooked. Firstly, before utilizing Pearson's correlation coefficient ($r$) to assess

the PRR, it is essential to ensure that certain assumptions are met, such as the bivariate normality of the data (Armstrong, 2019). With the influence of climate change, land conversion, and human water use, both precipitation and runoff time series may exhibit non-stationary characteristics (Liu et al., 2015; Zhang et al., 2011b), indicating that the data does not follow a normal distribution. Hence, the application of the correlation coefficient for detecting PRR in non-stationary conditions is constrained. Furthermore, climate change and anthropogenic activities interact in highly interconnected ways, affecting the water cycle at various temporal scales (Abbott et al., 2019). For instance, climate change disrupts water flow and storage patterns at long-term time scales (Huang et al., 2016; Haddeland et al., 2014), while anthropogenic influences, such as agriculture, urbanization expansion, and deforestation, disturb the land surface on multi-decadal timescales (Qiu et al., 2017; Hu et al., 2021). Water withdrawals in a catchment, including those for agricultural, industry, and domestic sectors, are regulated at different times and locations (Wada et al., 2011). Therefore, the PRR may undergo temporal changes in catchments exhibiting non-stationary behaviors. However, Pearson's correlation coefficient fails to capture the non-stationarity in the PRR (Huang et al., 2015).

Another common method used to characterize the PRR is the event runoff coefficient ($rc$), which quantifies the fraction of precipitation that is retained within different storage components, such as vegetation interception, soil moisture accumulation, and percolation into deeper layers (Tarasova et al., 2018). It plays a pivotal role in estimating the discharge from these storage compartments as event runoff in basins (Merz et al., 2006). However, it is worth noting the constraints of the runoff coefficient in describing the relationships between precipitation and runoff. (1) The runoff coefficient is calculated as the ratio between the volume of the runoff during a specific event and the corresponding precipitation (Savenije, 1996), the unknown antecedent soil moisture condition plays an important role. (2) The runoff coefficient captures the association between precipitation and runoff and may not directly reflect the influence of a specific factor on the PRR, such as the catchment's climatic and land-cover characteristics, including evaporation, vegetation, groundwater, and impermeable area. Instead, the runoff coefficient indicates the integrated characteristics of a catchment. (3) The estimation of the runoff coefficient often relies on simplifying assumptions (Feng et al., 2016), which can introduce uncertainties or inaccuracies in the calculations. Hydrological modeling is another prevalent approach in describing precipitation-runoff relationships. Continuous simulation of the PRR is accomplished through the application of data-driven hydrological models or process-based models with intricate model structures (Sikorska-Senoner and Quilty, 2021; Nayak et al., 2013). Data-driven models such as Artificial Neural Networks (ANN) and Linear Perturbation Model (LPM) rely on empirical analysis to produce corresponding outputs based on specific input data (Tanty and Desmukh, 2015; Nash and Barsi, 1983). These models are heavily dependent on the characteristics of existing data and provide a relatively vague description of the hydrological cycle processes, failing to explicitly reflect the impact of individual factors within these processes. On the other hand, process-based hydrological models such as TOPMODEL and SWAT can reveal more details and mechanisms of physical processes (Beven et al., 2021; Krysanova and White, 2015). However, these models involve complex modeling processes and require careful selection based on adequate prior knowledge (Reichl et al., 2009). Furthermore, many traditional hydrological models regionalize the PRR over a specific period in stationary climatic and catchment conditions, assuming minimal human interference (Yang et al., 2022; Westra et al., 2014). As a result, they may not be suitable for non-stationary hydrological processes. With the advancement of hydrological models, the impact of human activities is increasingly incorporated into the simulation of hydrological processes. Models such as WEP-L, WaterGAP, PCR-GLOBWB, and CWatM not only simulate natural hydrological cycles but also effectively incorporate the influence of anthropogenic factors into their simulations (Jia et al., 2006; Müller Schmied et al., 2021; Sutanudjaja et al., 2018; Burek et al., 2020). However, it is crucial to recognize that such hydrological models have high data requirements, limiting their application in regions lacking sufficient observational data (Clark et al., 2016). For instance, data on human activities are often difficult to obtain comprehensively, and while some remote sensing data may be

used, the impact of human activities on runoff represented by these data is complex and cannot be directly used as model input. Therefore, there is a need for a simple and effective PRR technique that has lower data requirements while being able to explore the impact of individual factors on PRR under non-stationary conditions.

Detrended Fluctuation Analysis (DFA) is a widely used technique in the investigation of power-law autocorrelations within non-stationary time series data (Peng et al., 1994). This approach offers significant advantages as it effectively removes the influence of polynomial trends present in the data and facilitates the identification of scale properties (Kantelhardt et al., 2001). Detrended Cross-Correlation Analysis (DCCA), derived from DFA, was initially proposed by Kantelhardt et al. (2002) as a method for detecting power-law cross-correlation between non-stationary signals. Trends in non-stationary time series can negatively affect cross-correlations identification (Kwiatkowski et al., 1992). Thereby, DCCA represents a modified approach to conventional covariance analysis of time series (Podobnik and Stanley, 2008). The Detrended Partial-Cross-Correlation Analysis (DPCCA), built upon DCCA and utilizing a partial-correlation technique, serves to mitigate the influence of trends within non-stationary time series when determining cross-correlations (Yuan et al., 2015). Furthermore, it effectively eliminates the impact of other signals on the two examined signals, revealing the intrinsic cross-correlations between them (Shen and Li, 2016; Zhang et al., 2020b). Precipitation and runoff time series may be stationary or non-stationary, and other influencing factors may also influence their relationships. This technique has the potential to deepen and broaden our understanding of the interactional mechanisms underlying the precipitation-runoff correlation in non-stationary scenarios. Given these, and developed from the benefits of DCCA and DPCCA for cross-correlation analysis in non-stationary situations, we propose an index to identify the potential effects of driving level and direction of one influencing factor in the changes of PRR, and address the limitations of DCCA and DPCCA. This index allows for a comparison of driving levels and directions without being constrained by inconsistent data-sequence lengths and across various types of catchments. Furthermore, the index provides a definitive measure of cross-correlation based on kernel density function. It facilitates an assessment of the temporal dynamics of driving levels and directions of changes in the PRR. Considering the linear characteristics of the index, it pre-processes the time lag between precipitation and the mass centers of baseflow. Furthermore, the index characterizes the uncertainty associated with driving levels in PRR changes across various time scales.

In these regards, the objective of this study is to investigate the potential processes controls for changes in precipitation-runoff relationships in non-stationary conditions and use the findings to gain an in-depth insight into hydrological processes, facilitating informed decisions for sustainable water resource management in a catchment. The study focuses on three main aspects:

- Developing an integrated framework for exploring the potential processes controls on changes in precipitation-runoff relationships in non-stationary environments.
- Proposing a novel Driving index for changes in Precipitation-Runoff Relationships (DPRR) to quantify the driving levels and directions of factors influencing precipitation-runoff links.
- Establishing a holistic conceptual model of catchment response, integrating DPRR values to infer the potential processes controlling changes in precipitation-runoff relationships.

To achieve these objectives, this study has selected five sub-basins with varying catchment characteristics within the Wei River Basin in Northern China. These sub-basins experience significant impacts of climate change and human activities, to serve to effectively illustrate our research objectives and methodologies.

## 2 Case study area and data description

### 2.1 Case study area

The Wei River Basin (103°05′E–110°05′E, 33°50′N–37°05′N) is the largest tributary of the Yellow River Basin in China (Zuo et al., 2015) (see Figure 1). It spans a wide range of climates and elevations, gradually decreasing from northwest to southeast, and covers a drainage area of 134,800 km² with a total length of 818 km (Huang et al., 2017b). The basin experiences a continental monsoon climate, and the dry/wet conditions have high spatiotemporal variability. The annual precipitation in the Wei River Basin is approximately 572 mm, mainly concentrated between June and October. The multi-year average temperature in the basin is around 10.6°C. The mean annual runoff is about 60 mm (Zhao et al., 2015). Especially, the intricate interplay of anthropogenic activities has led to a notable declining trend in the annual streamflow observed (Zhang et al., 2022). It implies that the assumption of stationary hydrologic properties has been challenged in the Wei River Basin (Xiong et al., 2018; Jiang et al., 2023). In light of this, the Wei River Basin is a suitable study area to analyze the precipitation-runoff dependency in a non-stationary hydrological system. Five sub-basins with different properties of precipitation-runoff dependencies and varying anthropogenic interventions in the Wei River Basin are applied as illustrations in this study. The five sub-basins (WR1, WR2, WR3, WR4, and WR5) are controlled by Qinan, Weijiabao, Xianyang, Zhangjiashan, and Zhuangtou hydrological monitoring stations (Figure 1), respectively.

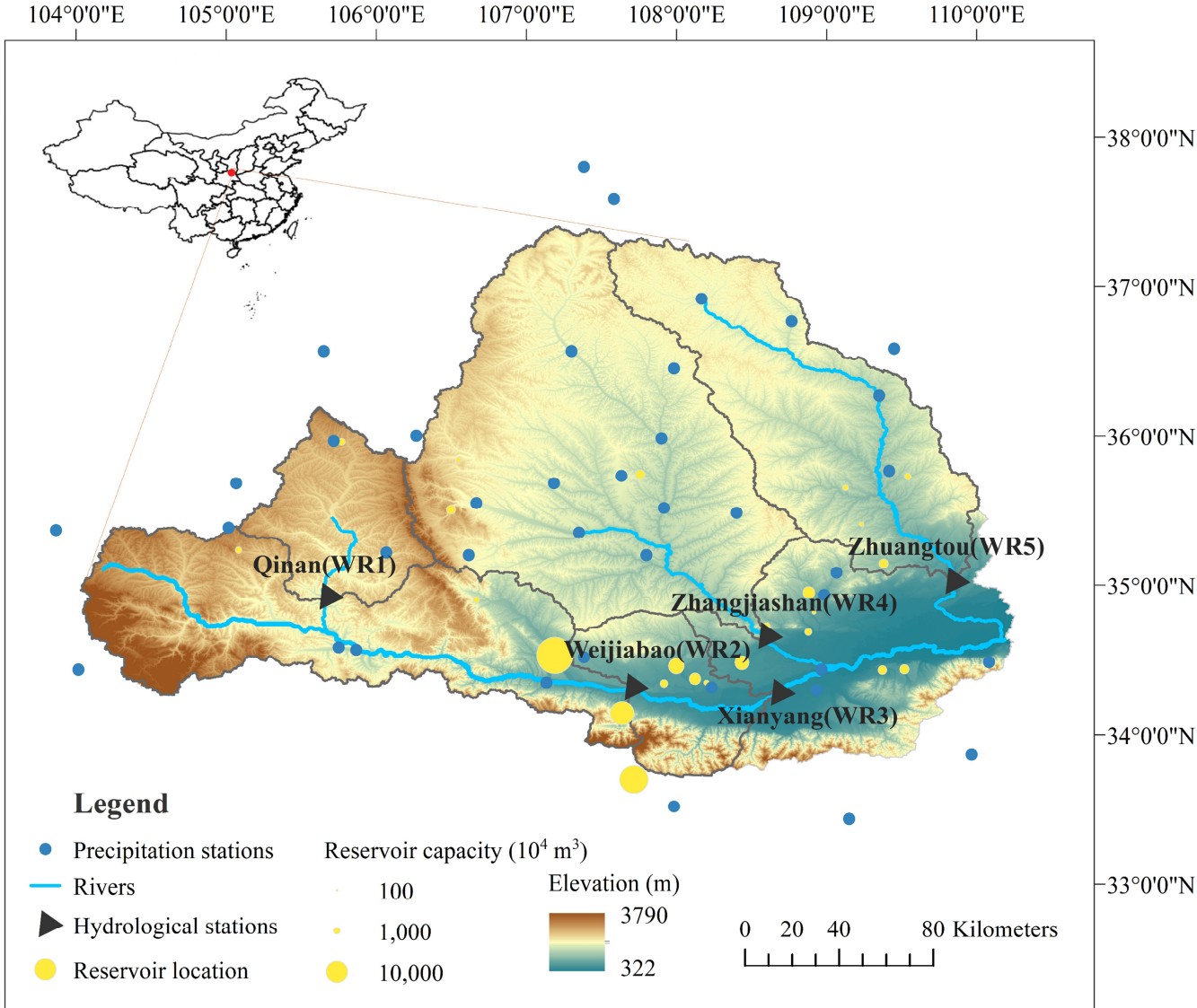

**Figure 1.** Information of the study area and five selected sub-basins (WR1, WR2, WR3, WR4, and WR5), hydrological monitoring stations, meteorological stations (precipitation stations), and primary reservoirs within the Wei River Basin. In the upper-left figure, the red circle indicates the geographical location of the Wei River Basin in China.

## 2.2 Data description

This study utilizes four main types of data: observed streamflow data, meteorological data, geographical information data, and remote sensing datasets related to anthropogenic influences. Daily streamflow data from the Yellow River Conservancy Commission were collected for five sub-basins and are reported in Table S1 of Supporting Information S1, which includes the drainage area, duration, longitude, and latitude of hydrological monitoring stations. Daily baseflow (*BF*) was calculated using the Chapman-Maxwell filter method, as detailed in Supporting Information S2. Given the critical role of baseflow in this study, the calculated values have been validated by relevant literature to ensure accuracy and reliability. Daily meteorological data from 1960-2019 were obtained from the China National Surface Weather Station (V3.0) (http://www.nmic.cn/), which involve precipitation, mean, maximum, and minimum air temperature, actual atmospheric pressure, wind speed, relative humidity, and sunshine duration. The quality of data from 39 meteorological stations (Figure 1 and Table S2) was rigorously controlled to ensure

high reliability before its release. Missing data, which were less than 0.01%, were reconstructed using linear regression techniques and neighboring stations. Potential Evapotranspiration ($ET_0$) was estimated using the Penman-Monteith equation (Supporting Information S2). A 90 m resolution Digital Elevation Model (DEM) was obtained from the USGS Shuttle Radar Topography Mission (SRTM) Digital Elevation Database (http://srtm.csi.cgiar.org/). The Global Inventory Modelling and Mapping Studies (GIMMS) NDVI3g dataset (https://ecocast.arc.nasa.gov/data/pub/gimms/3g.v1/) was used to analyze vegetation cover dynamics from 1982 to 2014 with a spatial resolution of 1/12°. Impervious Surface Ratio (ISR) data from 1985 to 2019 at a spatial resolution of 30 m were collected to infer the rate of urbanization (Gong et al., 2020). The ISR dataset was extracted and calculated from the Google Earth Engine platform. The Defense Meteorological Satellite Program's Operational Linescan System provides a free annual time series of night-time light (NTL) satellite images through the NOAA National Geophysical Data Center (Ngdc, 2013). The NTL data used in this study cover the period from 1992 to 2019 at a spatial resolution of 30-arc seconds. Population data (POP) from 2000 to 2019 was obtained from WorldPop (Worldpop, 2018), which is the data on population distributions and dynamics at the high spatial resolution, which characterizes population growth and rural-urban migration. The POP dataset was processed by an unconstrained top-down modeling method into a resolution of 30 arc seconds. The processing of NDVI data involves averaging the raster data within each study area. For ISR data, the number of impervious surface rasters within each study area is divided by the total number of rasters. NTL data is processed by summing the nighttime light intensity within each study area. Similarly, POP data is processed by summing the population within each study area. In addition, the regulations of reservoirs, as a significant anthropogenic intervention in the water cycle of the Wei River Basin, are further investigated. Information regarding the completion dates, storage capacities, and spatial distribution (Figure 1) of the 24 primary reservoirs has been obtained from the Yellow River Conservancy Commission. The annual water usage, including agricultural, industrial, and domestic sectors, in the Wei River Basin between 1956 and 2009 (obtained by Yellow River Conservancy Commission), was analysed to facilitate the assessment and validation of the anthropogenic influences on the precipitation-runoff links.

## 3 Methods

The methodology flow of this research is detailed in the Supporting Information S3.

### 3.1 Detection of non-stationary processes

The non-stationary hydrological processes in the study area were initially identified in this study. Mean and trend are essential statistical properties that remain constant over time in stationary conditions and are typically utilized to identify abrupt shifts and gradual changes in non-stationary conditions, respectively (Cryer and Kellet, 1991). To identify abrupt shifts in the hydrological time series involving runoff and baseflow, we employed the Pettitt test in conjunction with the Trend-Free Pre-Whitening and Binary Segmentation techniques (TFPW-BS-Pettitt). The Pettitt test is designed to detect changes in mean values (Mallakpour and Villarini, 2016), and the Trend-Free Pre-Whitening (TFPW) procedure (Yue et al., 2003) is applied in the Pettitt algorithm to mitigate the significant autocorrelation effects of non-stationary time series. The Binary Segmentation (BS) method (Lee and Verma, 2012) segments sub-periods prior to Pettitt's detection through iterative processes to identify multiple abrupt shift points in the time series. The non-parametric Mann-Kendall (MK) statistical test is used to assess trends or gradual changes in the time series (Yue and Wang, 2004), with significant sequential autocorrelation being removed using TFPW procedures before the MK test. For the detailed calculation process of TFPW-BS-Pettitt and TFPW-MK, please refer to Supporting Information S4.

**3.2 Identification of potential drivers of the changes in precipitation-runoff relationships**

The candidate driving factors for changes in PRR in non-stationary processes consider climate forcing, groundwater, vegetation dynamics, and anthropogenic influences. Furthermore, a holistic conceptual model of catchment response (Figure 2a) is developed by integrating the possible explanations (Figure 2b) that align with empirical evidence and logical reasoning. The potential driving processes of the candidate driving factors are elaborated in Supporting Information S5. Furthermore, a detailed description of this conceptual model is elaborated in the Discussion section based on the Results section, and validated by Information Theory and

existing literature findings.

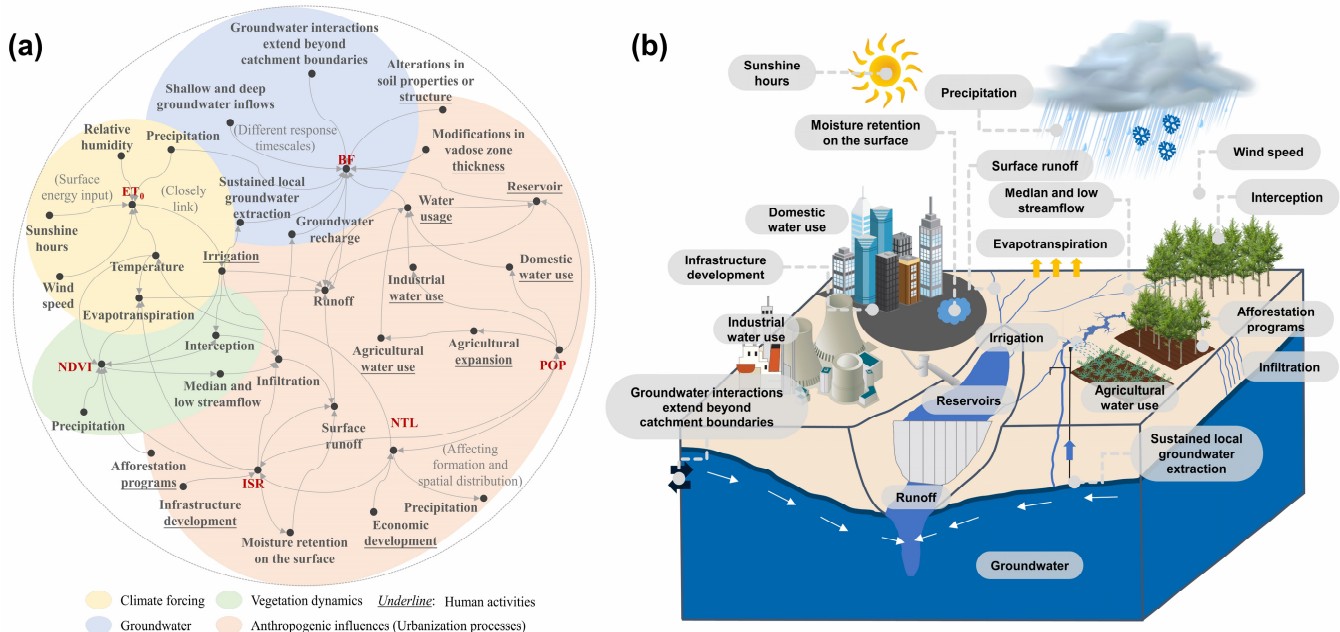

**Figure 2. a,** A conceptual model of catchment response developed by integrating the causal loop-based process explanations through investigating candidate driving factors. **b,** Visual synthesis of selected process explanations for potential driving mechanisms of the changes in PRR under non-stationary processes, depicting a general catchment affected by anthropogenic

interference.

A Driving index for changes in Precipitation-Runoff Relationships (DPRR) is proposed as a means to identify the potential driving mechanisms that influence the precipitation-runoff connections within non-stationary hydrological processes. The index effectively overcomes the limitations associated with conventional approaches used to describe PRR and their driving mechanisms in non-stationary conditions. The calculation procedure for the DPRR index is as follows.

**Step 1:** The time lag between precipitation and baseflow is given by,

$$R(m) = \sum_{i=1}^{N} x_i^{\text{Pre}} x_{i-m}^{\text{BF}} \tag{1}$$

The $R(m)$ is the cross-correlation coefficient, which is calculated by the XCORR function; $m$ is the time lag, and its range is [-12, 0], which was selected based on the monthly time scale; $x_i^{\text{Pre}}$ is precipitation time series ($i$ = 1, 2, 3, …, $N$); $x_{i-m}^{\text{BF}}$ represents the baseflow time series, where $x_{i-m}^{\text{BF}}$ is equal to zero when $i - m$ exceeds $N$.

$$R(m_{\text{max}}) = \max\{R(m)\} \tag{2}$$

where $m_{max}$ corresponds to the time lag $m$ at which $R(m)$ achieves its maximum value.

$$\begin{cases} x_i^{BF\_lag} = x_{i-m_{max}}^{BF} & i - m_{max} \leq N \\ x_i^{BF\_lag} = x_N^{BF} & i - m_{max} > N \end{cases} \tag{3}$$

where $x_i^{BF\_lag}$ denotes the baseflow time series at the time lag of $m_{max}$.

**Step 2:** Suppose that the precipitation time series, runoff time series, and the influencing factor of PRR are $\{x_i^1\}$, $\{x_i^2\}$, $\{x_i^3\}$ ($i = 1$, 2, 3, …, $N$), respectively. Each time series $\{x_i^j\}$ ($j = 1, 2, 3$) is accumulated as the profile $\{P_k^j\}$ ($k = 1, 2, 3, …, N$).

$$P_k^j = \sum_{i=1}^{k} x_i^j \tag{4}$$

**Step 3:** The profiles are divided into $N - s + 1$ overlapping sub-periods. The time nodes of each sub-period $h$ are from $h$ to $h + s - 1$ ($h = 1, …, N - s + 1$). Then, the local trend $\{\widetilde{P_{k,h}^j}\}$ is generated by a least-squares polynomial fitting. Accordingly, the 

detrended residual series $\{Y_{s,l}^j\}$ ($l = (h-1) \cdot s + k - h + 1$) are calculated by the difference between the original time series and the local trend.

$$Y_{s,l}^j = P_k^j - \widetilde{P_{k,h}^j} \tag{5}$$

**Step 4:** The cross-correlation levels $\rho_{j_1,j_2}(s)$ between any two-time series on the time scales of $s$ are estimated, which ranges from -1 to 1. The $\rho_{j_1,j_2}(s)$ is also referred to as the DCCA (Detrended Cross-Correlation Analysis) index, which characterizes the PRR in non-stationary hydrological processes. The coefficients matrix is constituted as,

$$\rho_{j_1,j_2}(s) = \frac{\sum_{l=1}^{(N-s+1) \cdot s} Y_{s,l}^{j_1} Y_{s,l}^{j_2}}{\sqrt{\sum_{l=1}^{(N-s+1) \cdot s} Y_{s,l}^{j_1} Y_{s,l}^{j_1}} \sqrt{\sum_{l=1}^{(N-s+1) \cdot s} Y_{s,l}^{j_2} Y_{s,l}^{j_2}}} \tag{6}$$

$$\rho(s) = \begin{bmatrix} \rho_{1,1}(s) & \rho_{1,2}(s) & \rho_{1,3}(s) \\ \rho_{2,1}(s) & \rho_{2,2}(s) & \rho_{2,3}(s) \\ \rho_{3,1}(s) & \rho_{3,2}(s) & \rho_{3,3}(s) \end{bmatrix} \tag{7}$$

**Step 5:** The partial-cross-correlation level $\rho(1,2;s)$ between the precipitation and runoff is determined based on the inverse matrix of $\rho(s)$ which is defined as,

$$C(s) = \rho^{-1}(s) = \begin{bmatrix} C_{1,1}(s) & C_{1,2}(s) & C_{1,3}(s) \\ C_{2,1}(s) & C_{2,2}(s) & C_{2,3}(s) \\ C_{3,1}(s) & C_{3,2}(s) & C_{3,3}(s) \end{bmatrix} \tag{8}$$

$$\rho(1,2;s) = \frac{-C_{1,2}(s)}{\sqrt{C_{1,1}(s) C_{2,2}(s)}} \tag{9}$$

**Step 6:** The $\rho_{1,2}(s)$ denotes the PRR on the time scales of $s$ with removing the non-stationary effects. The $\rho(1,2;s)$ characterizes the cross-correlation between precipitation and runoff by eliminating the influences of an external factor. The difference between $\rho_{1,2}(s)$ and $\rho(1,2;s)$ represents the driving level of an influencing factor on PRR. To address the issue of inconsistent data-

sequence lengths among different driving factors within a catchment and facilitate comparisons between different catchments, we introduce the concept of relative error. It is computed by dividing the difference between $\rho_{1,2}(s)$ and $\rho(1,2;s)$ by $\rho_{1,2}(s)$. The $\rho_{1,2}(s)$ represents the specific PRR of a catchment at a given time scale $s$, reflecting the catchment-properties PRR influenced by

integrated multifactorial forces. However, when the denominator $\rho_{1,2}(s)$ approaches zero, there is a risk of encountering abnormally high values in the DPRR index. To address this, we update the denominator to $\rho_{1,2}(s) + 1$. Notably, a positive value of the DPRR index signifies that the driving factor positively enhances the P-R link, while a negative index value suggests a negative effect of the driving factor on the P-R link. In this context, the calculation of the DPRR index is given by,

$$\text{DPRR}(1,2;3;s) = \frac{\rho_{1,2}(s) - \rho(1,2;s)}{\rho_{1,2}(s) + 1} \tag{10}$$

**Step 7:** The above formula for DPRR can only provide index values for different time scales. However, we prefer a definitive measure within a given period. Hence, the kernel density function is utilized.

$$\hat{f}_w(\text{DPRR}) = \frac{1}{(b - a + 1)w} \sum_{s=a}^{b} K\left(\frac{\text{DPRR} - \text{DPRR}(1,2;3;s)}{w}\right) \tag{11}$$

$$\hat{f}_w(\text{DPRR}_{\max}) = \max\{\hat{f}_w(\text{DPRR})\} \tag{12}$$

where $\hat{f}_w(\text{DPRR})$ represents the kernel density function with $w$ which denotes the bandwidth. The range of $s$ is set as $[a, b]$, and a GridSearchCV is conducted for the optimal range (Pedregosa et al., 2011). The search space for $w$ is set to $[0.05, 1.95]$ with a step size of 0.05. $K(.)$ denotes the kernel function. $\text{DPRR}_{\max}$ corresponds to the maximum value of kernel density for DPRR. Here, the Gaussian kernel density function is utilized for estimating kernel density.

$$K(x) = \frac{1}{\sqrt{2\pi}} \exp\left(-\frac{1}{2} x^2\right) \tag{13}$$

The kernel density estimations for $\rho_{1,2}$ and $\rho(1,2)$ in DPRR are also computed.

$$\hat{f}_w(\rho_{1,2}) = \frac{1}{(b - a + 1)w} \sum_{s=a}^{b} K\left(\frac{\rho_{1,2} - \rho_{1,2}(s)}{w}\right) \tag{14}$$

$$\hat{f}_w(\rho_{1,2;\max}) = \max\{\hat{f}_w(\rho_{1,2})\} \tag{15}$$

$$\hat{f}_w(\rho(1,2)) = \frac{1}{(b - a + 1)w} \sum_{s=a}^{b} K\left(\frac{\rho(1,2) - \rho(1,2;s)}{w}\right) \tag{16}$$

$$\hat{f}_w(\rho(1,2)_{\max}) = \max\{\hat{f}_w(\rho(1,2))\} \tag{17}$$

A Dynamic Driving index for changes in Precipitation-Runoff Relationships (D-DPRR) is further designed to evaluate the temporal dynamics in the driving level and direction of the influencing factors on the PRR in non-stationary conditions.

**Step 1:** For the D-DPRR, the definition and accumulation of the original time series in the first step are consistent with DPRR.

**Step 2:** The cumulative time series are synchronously divided into $N - s + 1$ overlapping sub-periods, and their start and end time nodes are from $h$ to $h + s - 1$ ($h = 1, \ldots, N - s + 1$). Then, the local trend $\{\widetilde{P_{k,h}^J}\}$ is removed in each sub-period $h$ and the detrended residual series $\{Y_{s,h,u}^j\}$ are refined considering the $u^{\text{th}}$ ($u = k - h + 1$) element in each sub-period.

$$Y_{s,h,u}^j = P_k^j - \widetilde{P_{k,h}^J} \tag{18}$$

**Step 3:** The cross-correlation levels $\rho_{j_1,j_2}(s;h)$ and the coefficients matrix in each sub-period $h$ is defined as,

$$\rho_{j_1,j_2}(s;h) = \frac{\sum_{u=1}^{s} Y_{s,h,u}^{j_1} Y_{s,h,u}^{j_2}}{\sqrt{\sum_{u=1}^{s} Y_{s,h,u}^{j_1} Y_{s,h,u}^{j_1}} \sqrt{\sum_{u=1}^{s} Y_{s,h,u}^{j_2} Y_{s,h,u}^{j_2}}} \tag{19}$$

$$\rho(s;h) = \begin{bmatrix} \rho_{1,1}(s;h) & \rho_{1,2}(s;h) & \rho_{1,3}(s;h) \\ \rho_{2,1}(s;h) & \rho_{2,2}(s;h) & \rho_{2,3}(s;h) \\ \rho_{3,1}(s;h) & \rho_{3,2}(s;h) & \rho_{3,3}(s;h) \end{bmatrix} \tag{20}$$

**Step 4:** The partial-cross-correlation level $\rho(1,2;s;h)$ between precipitation and runoff in each sub-period is constructed based on the inverse matrix $\rho^{-1}(s;h)$ as follows,

$$C(s;h) = \rho^{-1}(s;h) = \begin{bmatrix} C_{1,1}(s;h) & C_{1,2}(s;h) & C_{1,3}(s;h) \\ C_{2,1}(s;h) & C_{2,2}(s;h) & C_{2,3}(s;h) \\ C_{3,1}(s;h) & C_{3,2}(s;h) & C_{3,3}(s;h) \end{bmatrix} \tag{21}$$

$$\rho(1,2;s;h) = \frac{-C_{1,2}(s;h)}{\sqrt{C_{1,1}(s;h)C_{2,2}(s;h)}} \tag{22}$$

**Step 5:** Similar to DPRR, the D-DPRR is determined by cross-correlation levels $\rho_{j_1,j_2}(s;h)$ and the partial-cross-correlation level $\rho(1,2;s;h)$ in each period $h$.

$$\text{D-DPRR}(1,2;3;s;h) = \frac{\rho_{1,2}(s;h) - \rho(1,2;s;h)}{\rho_{1,2}(s;h) + 1} \tag{23}$$

Compared to conventional hydrological models, (1) the DPRR index has lower data requirements and offers a simple and effective technique for identifying the potential impacts of driving factors on PRR. (2) It also addresses the limitations of process-driven hydrological models in the run, which assume stationary conditions (Ammann et al., 2019; Jehanzaib et al., 2020). Hence, the proposed index offers crucial information for the hydrological cycle process driven by climate change or anthropogenic disturbances, which guides the construction of more robust hydrological models and the development of water resource management and allocation. (3) Considering the elimination of trends is a crucial step in accurately analyzing the relationships between complex non-stationary systems (Zhao et al., 2012), the DPRR removes the non-stationary effects by subtracting local trends with appropriate polynomial orders, ensuring the normality of input signals for cross-correlation analysis (Zebende, 2011). (4) The effect of external factors on PRR may lead to spurious cross-correlation estimations (Yuan et al., 2015). Hence, the developed index applies the theoretical foundations of the DPCCA technique to reveal the "intrinsic" relations between precipitation and runoff time series with potential influences of other factors removed, such as evapotranspiration, groundwater, land cover, and anthropogenic interference. (5) The DPRR index characterizes the potential driving mechanisms influencing PRR on different time scales, which can improve our understanding of hydrological responses to climate forcing and anthropogenic activities at various time scales. Within a specified period, the driving level of DPRR signifies the influence level exerted by a particular factor on the correlation between precipitation and runoff during the period, and the driving direction of DPRR signifies whether a particular factor has positive or negative effects on the PRR during the period. (6) The DPRR index provides the driving level and direction and allows for comparisons of the index values among different driving factors with inconsistent data-sequence lengths and across various types of catchments. (7) Indeed, while DCCA and DPCCA can only capture the PRR at different time scales, they do not provide a definitive measure of cross-correlation (Yuan et al., 2015). Therefore, the kernel density function is applied to the DPRR index to provide a definitive value for exploring the potential processes controls of PRR. (8) The Dynamic DPRR (D-DPRR) is proposed as a metric to assess the temporal dynamics of driving levels and directions of changes in the PRR

within non-stationary hydrological processes. (9) Baseflow, which plays a crucial role in the PRR, is subjected to a pre-processing step involving the determination of the time lag between precipitation and the mass centers of baseflow (Singh, 1968). This pre-processing step is performed prior to the application of the DPRR, allowing for a more accurate analysis of the potential drivers of the changes in PRR. (10) The uncertainty associated with the driving levels on the changes of PRR across various time scales is characterized by violin plots. The violin plot combines the box plot and density plot to provide a detailed representation of data distribution. The wider sections of the violin plot indicate a higher probability of data distribution, whereas the narrower sections suggest a lower probability (Hintze and Nelson, 1998). Therefore, given the same volume of data, a vertically flatter or multimodal violin plot signifies a lower concentration and higher uncertainty of driving levels for the changes in the PRR as the time scale changes.

## 4 Results

### 4.1 Non-stationarity of hydrological processes

The non-stationary testing results for hydrological process-related datasets are presented in Table 1. The significant probability $q(t)$ values derived from the TFPW-BS-Pettitt method were utilized to identify the most significant abrupt time points of runoff in WR1, WR2, WR3, WR4, and WR5, which were found to occur in 1992, 1993, 2002, 1996, and 1994, respectively. Figure 3 depicts the subordinate significantly abrupt shift points of the runoff time series. The significantly abrupt shift points of the baseflow were found to be consistent with the runoff time series in the five sub-basins. The potential reason is that the Wei River Basin is situated in a semi-arid region where streamflow is primarily influenced by groundwater (Zhao et al., 2015).

Trends in the precipitation time series across the five sub-basins are non-significant at a 5% significant level using the TFPW-MK detection approach. However, the temperature time series exhibited a significant upward trend. Also, the potential evapotranspiration series shows a significant increasing trend in WR2 and WR3, while a non-significant increasing trend was observed in the remaining sub-basins. An exploratory analysis was conducted to investigate the potential mechanism for significant or non-significant non-stationary changes in climate time series within the Wei River Basin. The global warming scenarios affecting the region have resulted in a sustained increase in the long-term temperature time series of five sub-basins (Zuo et al., 2015). Furthermore, the temperature variations in the sub-basins are also influenced by other regional factors, such as urban expansion, which can exacerbate the temperature rise (Huang et al., 2021). The persistent stationarity observed in the precipitation time series within the Wei River Basin was attributed to the integrated effect of climate forcing associated with the reorganization of global-scale and regional-scale climate processes, as well as the movement of large-scale atmospheric circulation system (Saft et al., 2015). The NDVI values manifest a significantly upward trend on an annual time scale in the five sub-basins. This phenomenon is linked to the execution of the soil and water conservation project (large-scale afforestation and reforestation) in the Wei River Basin since the 1950s, with extensive implementation in the 1970s (Chen et al., 2007). Other satellite data representing the human pressure on the river system of the Wei River Basin, including ISR, NTL, and POP, overall exhibit a significant upward trend besides the population in WR1 and WR4. This is attributed to the expansion of urbanization and the intensification of human activities on the water resource system in the Wei River Basin over the past few decades (Chang et al., 2015).

**Table 1.** Non-stationary analysis and trend test results for hydrological process datasets (q is the significant probability of non-stationary analysis, a positive value of Z means that the series has an upward trend, a negative value of Z means that the series has a downward trend, Y means that the trend is significant, and N means that the trend is not significant).

| Stations | R | | P | | T | | $ET_0$ | | BF | | NDVI | | ISR | | NTL | | POP | |
|---|---|---|---|---|---|---|---|---|---|---|---|---|---|---|---|---|---|---|
| | q | Abrupt shifts | Z | Sig. | Z | Sig. | Z | Sig. | q | Abrupt shifts | Z | Sig. | Z | Sig. | Z | Sig. | Z | Sig. |
| WR1 | 0.99 | 1992 | -1.19 | N | 4.58 | Y | 0.20 | N | 0.99 | 1992 | 4.11 | Y | 7.98 | Y | 6.53 | Y | -5.89 | Y |
| | 0.97 | 1970 | | | | | | | 0.98 | 1970 | | | | | | | | |
| WR2 | 0.99 | 1993 | -0.90 | N | 5.24 | Y | 3.54 | Y | 0.99 | 1993 | 4.17 | Y | 8.04 | Y | 6.74 | Y | 2.39 | Y |
| | 0.97 | 1970 | | | | | | | 0.97 | 1970 | | | | | | | | |
| WR3 | 0.99 | 2002 | | | | | | | 0.99 | 2002 | | | | | | | | |
| | 0.96 | 1993 | -0.90 | N | 5.17 | Y | 2.26 | Y | 0.96 | 1993 | 4.39 | Y | 8.07 | Y | 6.74 | Y | 4.74 | Y |
| | 0.94 | 1968 | | | | | | | 0.95 | 1968 | | | | | | | | |
| WR4 | 0.97 | 1996 | -0.02 | N | 6.04 | Y | 1.56 | N | 0.98 | 1996 | 3.97 | Y | 7.89 | Y | 6.96 | Y | -5.34 | Y |
| WR5 | 0.91 | 1994 | -1.40 | N | 4.31 | Y | -0.22 | N | 0.85 | 1994 | 2.03 | Y | 7.73 | Y | 6.82 | Y | 4.20 | Y |

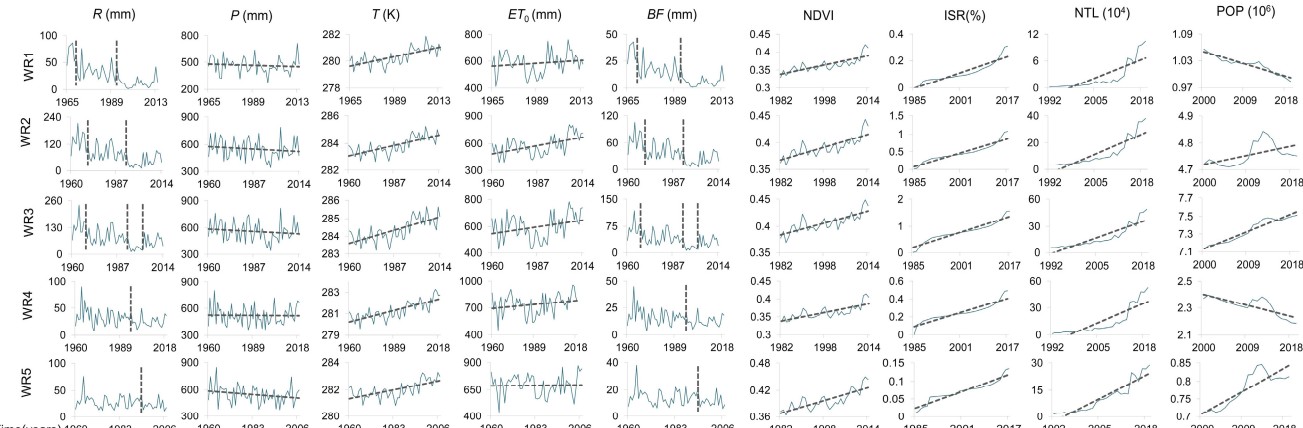

**Figure 3.** Time series and their abrupt shifts or gradual trends of interest variables.

## 4.2 Changes in the precipitation-runoff relationships

The monthly precipitation-runoff relationships of five sub-basins within the Wei River Basin were examined in non-stationary hydrological processes using Detrended Cross-Correlation Analysis (DCCA) (see *Eq*. (5)). The results of DCCA are presented in Figure 4a. It is evident that WR2 and WR3 exhibit the highest PRR values, concentrated around 0.8 and 0.87, respectively. These two sub-basins demonstrate lower levels of uncertainty when compared to the other sub-basins at different time scales. WR1 and WR4 display lower PRR values, primarily centered around 0.6, with higher uncertainties across different time scales. WR5 exhibits the lowest PRR values, around 0.5, showing the highest uncertainty when time scales vary. WR1 represents an upstream tributary with minimal anthropogenic impacts, while WR4 and WR5 are downstream tributaries that are situated within the semi-arid Loess Plateau. The heat map in Figure 4b illustrates the temporal variations in the PRR, providing insights into the dynamic hydrological response of the PRR over time, according to the non-stationarity detection results (refer to Section 4.1). WR2 and WR3 demonstrate stable PRR values even during non-stationarity in the hydrological processes. In WR1, noticeable low values (close to 0) of PRR

are observed in the mid-time scales before and after the abrupt hydrological shifts (e.g., in 1992 and 1970), indicating a deteriorated PRR with the occurrence of hydrological non-stationarity. Similarly, WR4 exhibits weaker precipitation-runoff links in the mid-time scales preceding the abrupt hydrological shifts. WR5 shows poor PRR throughout the large and mid-time scales of the hydrological processes. The possible explanations are given in Section 5.1.

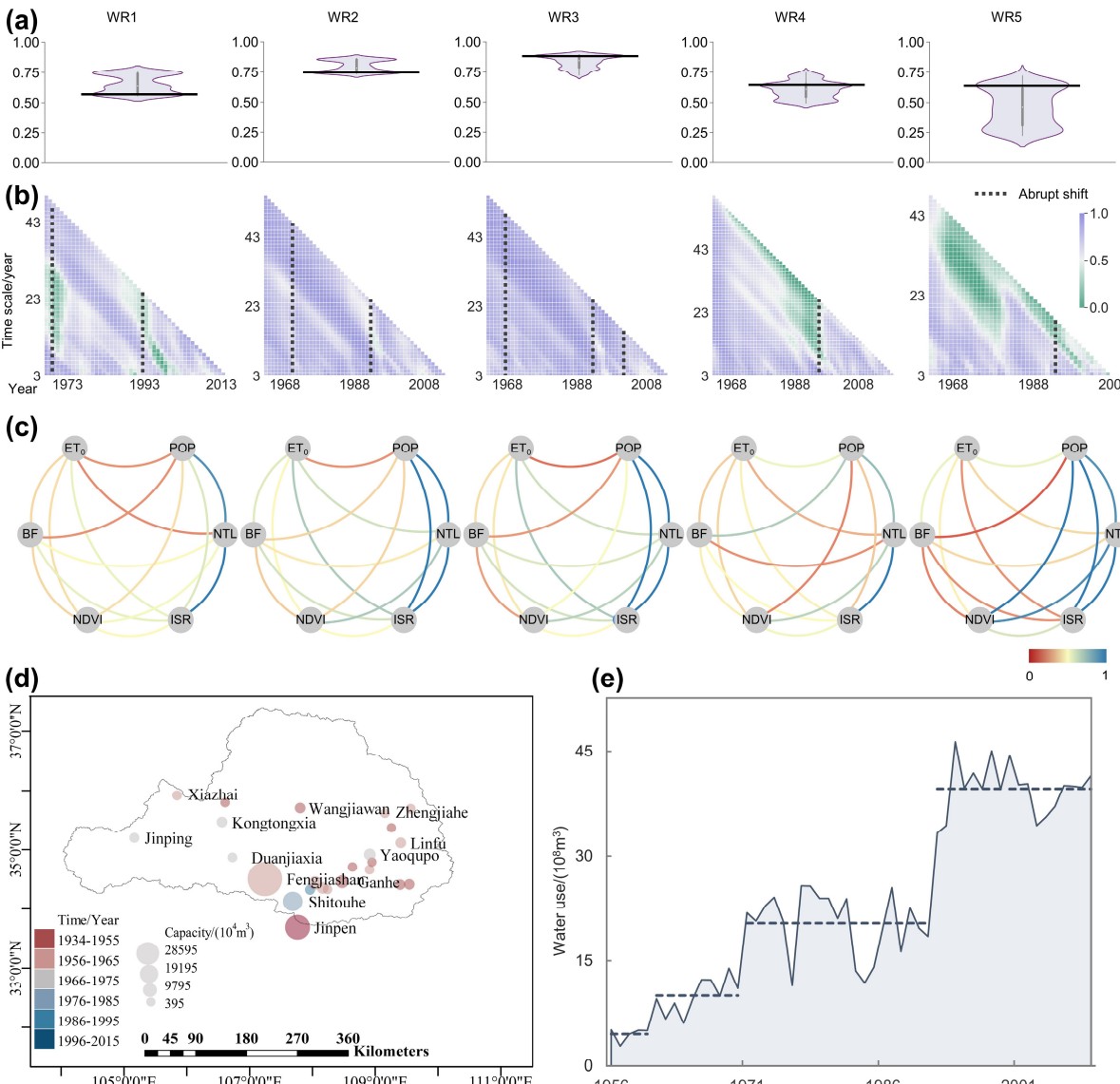

**Figure 4. a,** Precipitation-runoff relationships of five sub-basins within the Wei River Basin in non-stationary environments, investigated using DCCA values. **b,** DCCA values at various periods and time scales in the five basins (temporal variations in the PRR and the dynamic hydrological response of the PRR over time, examined through DCCA values). **c,** Interconnected network of the nonlinear relationships (Maximal Information Coefficient, MIC values) among the candidate driving factors. The edges correspond to the MIC values occurring for any two variables (minimum is red, maximum is blue). **d,** Completion time and storage capacities of main reservoirs in the Wei River Basin. e, Total water use in the Wei River Basin, including agricultural, industrial, and domestic water usage.

## 4.3 Driving levels and directions of potential influencing factors in changes of precipitation-runoff relationships

The driving levels and directions of potential influencing factors in precipitation-runoff relationships are quantified using DPRR, which are illustrated through the implementation of violin plots with maximum kernel density value (Figure 5a). To facilitate the comparison of the magnitudes of driving forces, the absolute values of DPRR are further depicted in Figure 5b. The results illustrate that baseflow is the primary driving force positively influencing the PRR in four out of the five sub-basins, excluding WR5. This finding aligns with the dominance of groundwater in the hydrological processes of the Wei River Basin (Miao et al., 2020). However, the influence of baseflow on the PRR exhibits significant uncertainty across various time scales. In WR5, the dominant factor affecting the PRR is potential evapotranspiration, which negatively impacts (or reduces) the precipitation-runoff dependency. Conversely, the influence of potential evapotranspiration on the PRR is small in other sub-basins, with less uncertainty in driving forces across different time scales. Vegetation dynamics negatively affect the PRR in all five sub-basins, ranking second in impact after baseflow in WR1, WR2, and WR3, with lower uncertainty compared to baseflow. In WR4, although the influence of vegetation dynamics on the PRR is small, it exhibits higher uncertainty. Notably, the impact of vegetation dynamics on the relationships is smaller in WR4 compared to WR5, which is consistent with Wu et al. (2023). Impermeability (ISR) has a stronger direct influence on the PRR compared to the other two indicators of urbanization, such as NTL and POP, and weakens the precipitation-runoff dependency in all sub-basins. Furthermore, their direction (positive or negative) is inconsistent, as their effects on the catchment's hydrological cycle are indirect and complex. The impact of NTL and POP on the PRR remains stable across different time scales. When comparing WR3 and WR4, as the urban cluster expands along the main stem of the Wei River Basin, the three indicators of anthropogenic influences, including ISR, NTL, and POP, have a greater driving effect on the PRR in WR4 compared to WR3 with smaller urban agglomerations. More detailed explanations of these results are provided in Section 5.1.

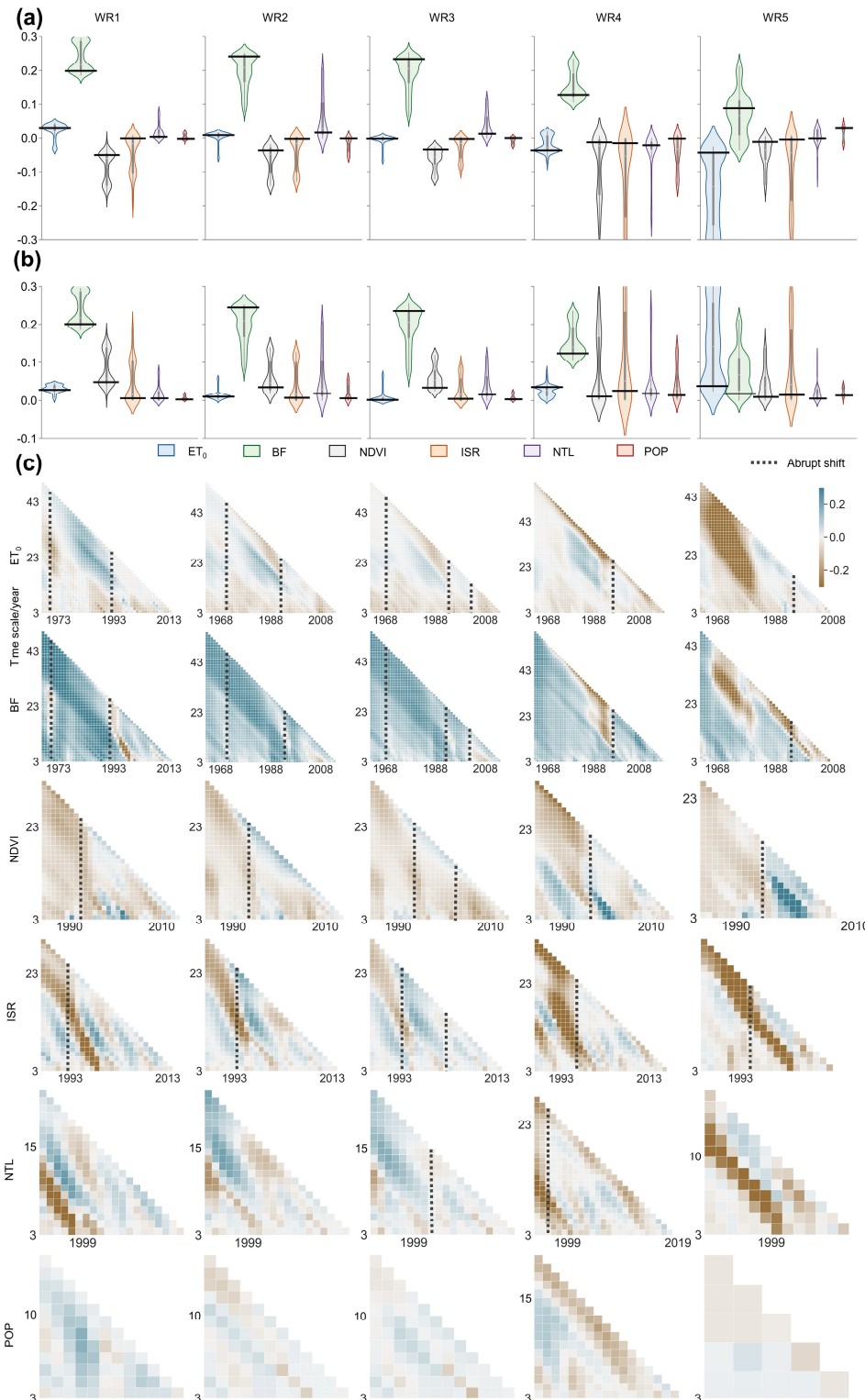

**Figure 5. a,** Driving levels and directions of potential influencing factors in changes of precipitation-runoff relationships within the Wei River Basin, as indicated by DPRR values. **b,** Absolute values of DPRR of potential influencing factors. **c,** Hydrological response to temporal dynamics of the driving factors based on the D-DPRR values.

### 4.4 Hydrological response to temporal dynamics of the driving factors

According to Figure 5c, in WR1, baseflow initially exhibits negative driving at the small-time scales (3-13 years) after the hydrological regime shift in 1992, followed by a lack of driving force at the medium-time scales before 1970. WR2 and WR3 show similar baseflow variability, with a transient negative driving at the small-time scale around 2007. In WR4, baseflow significantly contributes to a negative driving force on the PRR at the medium time scale prior to the hydrological shift point. In WR5, baseflow variations overall exhibit a negative driving force at both large and medium time scales. The impact of potential evapotranspiration on the PRR is little in WR1-WR4, remaining constant over time. However, in WR5, potential evapotranspiration shows a stronger negative driving force at the large and medium time scales before 1980. Vegetation dynamics have a negligible negative influence on the precipitation-runoff dependence in WR1, WR2, and WR3, with insignificant temporal changes. In WR4, vegetation dynamics exhibit a more significantly positive driving at the medium and small-time scales before and after the hydrological regime shift (1996). WR5 shows a significant positive driving force at the medium and small scales following the regime shift (1994). Overall, ISR, representing anthropogenic influence, predominantly exerts a negative impact on the PRR. However, ISR irregularly exhibits slight positive driving forces at the medium time scale, which are not well-aligned with the hydrological regime shifts. NTL and POP have a smaller overall driving force compared to ISR in the five sub-basins. However, due to the limitations of the time series length, they do not correspond well with the hydrological regime shifts. Possible process explanations for these results are provided in Section 5.1.

### 4.5 Nonlinear and intricate interplay among driving factors

#### 4.5.1 Nonlinear correlation among driving factors

Maximal Information Coefficient (MIC) metric provides a measure of both linear and nonlinear correlations among driving factors (Zhang et al., 2014), as shown in Figure 4c. The interconnected network illustrates that the three driving factors (POP, NTL, ISR) related to anthropogenic influences in five sub-basins exhibit significant linear or nonlinear correlations. However, the correlation between POP and ISR in WR1 and WR4 is weak. This was attributed to the migration of rural populations to urban regions in these two sub-basins (refer to Figure 3). Baseflow exhibits weak correlations with other factors. Moreover, potential evapotranspiration displays relatively weak correlations with other factors, likely due to the integrated effect of climate forcing associated with the reorganization of global-scale and regional-scale climate processes. In WR5, NDVI demonstrates strong associations with POP, NTL, and ISR, which may be attributed to the simultaneous implementation of soil and water conservation projects and urbanization processes in this sub-basin. For detailed information on MIC, please refer to Supporting Information S6.

#### 4.5.2 Other anthropogenic influences

The study investigates further the significant anthropogenic interference, specifically the regulations imposed on reservoirs and their impact on the PRR in the Wei River Basin, due to large-scale surface water withdrawals from the reservoirs primarily employed for irrigation in agricultural fields (Zhan et al., 2014). Notably, the main reservoirs in the Wei River Basin are commonly associated with corresponding irrigation districts. The size of these districts is correlated with the scale of the reservoirs. Therefore, the construction of reservoirs not only redistributes seasonal water discharge within a given year but also substantially modifies inter-annual distribution. However, the collection of data on reservoir storage variations presented challenges in this study. Therefore, the study primarily relies on the construction dates and storage capacities of the reservoirs (depicted in Figure 4d) to assess their impact on the PRR. In WR1, the Xiazhai Reservoir, with a small-scale capacity, was built in the 1970s. In WR2, the

largest reservoir, Fengjiashan Reservoir, was constructed in 1974. A significant concentration of reservoirs with large capacities is observed between WR2 and WR3 in the Wei River Basin. These include Zhangjiazuitou Reservoir, Shitouhe Reservoir, Xinyigou Reservoir, Yangmaowan Reservoir, Dabeigou Reservoir, Laoyaju Reservoir, and Shibianyu Reservoir. Based on the colors of the bubble chart, most of them were built simultaneously in the early 1970s. In WR4, the Xijiao Reservoir, with a large-scale irrigation district, was constructed in 1997. It directly controls the downstream streamflow volume at the Zhangjiashan gauge. In WR5, Zhengjiahe Reservoir and Linfu Reservoir, with minor capacities, were built in the 1970s. In addition, a large-scale irrigation area of Shibaochuan Reservoir is located downstream of WR5, regulating the streamflow at the Zhuangtou hydrological station. As a result, the timing of interference from the main reservoir regulations in the study cases closely aligns with hydrological regime shifts.

Analysis of the comprehensive water use data in the Wei River Basin (Figure 4e) reveals a significant trend. Since the 1950s, the total water usage in the basin, including agricultural, industrial, and domestic sectors, has shown a distinctive stair-step pattern of increase. This observed trend is consistent with the rapid urbanization process and the intensification of anthropogenic activities in the Wei River Basin (Chang et al., 2015).

## 5 Discussion

### 5.1 Potential processes controls

Figure 6 illustrates the causal loop-based process explanations derived from the investigation results of the candidate driving factors within the five sub-basins of the Wei River Basin. Using WR3 as a representative case, the changes in potential processes controls under non-stationary conditions are further elucidated. It is important to highlight that the causal loop-based process explanations are generally applicable to other catchments, as the generalized conceptual model is illustrated in Figure 2.

#### 5.1.1 Climate forcing

Potential evapotranspiration ($ET_0$) provides valuable insights into the integrated impact of meteorological factors on the PRR. Temperature, in particular, serves as a direct indicator of evaporation dynamics and exerts a primary influence on the magnitude of $ET_0$. In the five sub-basins, significant temperature increases have been observed (Figure 3). These increases may be attributed to the concurrent rise in sunshine hours, resulting in increased surface energy input that promotes evapotranspiration. Higher wind speed and temperature facilitate the movement of water molecules, favoring the transition of water from liquid to gaseous state. Conversely, high relative humidity limits evaporation by increasing the moisture content in the air. Despite the significant increase in temperature across the five sub-basins, the heterogeneous characteristics within each sub-basin led to diverse effects of $ET_0$ on PRR (Figure 5).

The relationships between $ET_0$ and the management and irrigation practices in irrigated areas are closely intertwined. Large-scale irrigation areas have been established in WR2, WR3, and WR5 (Figure 4d). $ET_0$ is associated with irrigation management (Tu et al., 2023; Berghuijs et al., 2017), affecting processes such as groundwater extraction, river water abstraction, vegetation cover, and infiltration, thereby producing complex effects on the PRR. Rapid urbanization in the regions WR2, WR3, and WR4, such as the construction of high-rise buildings, affects regional wind speed and wind direction, thus altering the effect pattern of $ET_0$ on PRR. In WR5, the influence of $ET_0$ on the PRR exhibits significant variations at different time scales (Figure 5c), likely attributable to the impact of large-scale afforestation on the local climate, subsequently affecting the PRR of sub-basin (Mu et al., 2007). The

observed variations indicate a complex process, which may be influenced by various factors related to the alteration of land cover and land use patterns resulting from afforestation.

### 5.1.2 Groundwater

According to the identification results of driving levels and directions, it was observed that baseflow had the most significant influence on the PRR compared to other factors. In WR5, *BF* was identified as the second most influencing factor in the PRR, closely following potential evapotranspiration. Five sub-basins are situated in the Loess Plateau, which falls in the continental monsoon climate region and experiences uneven distribution of precipitation throughout the year. The study area is characterized by high temperatures and abundant rainfall in summers, while winters are cold and dry. Baseflow contributes to runoff consistently

throughout the year. During the dry season with minimal or no precipitation, the low soil moisture content leads to a significant portion of the precipitation being absorbed by the soil or stored in water storage facilities within the basin, resulting in minimal surface runoff. At this time, baseflow becomes the primary contributor to runoff. As precipitation increases, soil moisture approaches saturation, and surface runoff generated by infiltration becomes the primary contributor to runoff, gradually reducing the contribution of baseflow (Miao et al., 2020).

The contribution of baseflow to runoff varies in different regions due to differences in groundwater depth (Huang et al., 2020). The results from the DPRR analysis of WR4 and WR5 (Figure 5) indicate that baseflow exhibits significant uncertainty at different time scales, which is more evident in the D-DPRR results. The D-DPRR analysis of WR4 and WR5 reveals significant differences between large and minor time scales, with a prominent negative response at long time scales. These findings suggest that deep groundwater has a greater impact on the PRR in these two sub-basins, and there are potentially negative effects on the PRR due to

anthropogenic activities such as groundwater extraction. The impact of baseflow on the PRR in the other three sub-basins (WR1, WR2, and WR3) also displays some uncertainty with respect to time scales, although it is not statistically significant.

     The groundwater system is closely interconnected and is not confined by surface topography, leading to frequent groundwater exchange between adjacent basins. Furthermore, human activities such as increased domestic, agricultural, and industrial water use can affect baseflow generation and thus influence the PRR (Huang et al., 2020). WR2 and WR3 flow through the largest urban

cluster centered in the Wei River Basin. They have high levels of economic development, resulting in rapid growth in water consumption and increased groundwater extraction. In this urban cluster, excessive deep-well pumping caused the confined water level to drop by approximately 100 meters until 1988 (Huang et al., 2017a). Additionally, with urbanization processes and population growth, the agricultural land area within the basin has increased, leading to a higher demand for irrigation water. WR2, WR3, and WR5 support massive irrigation projects, such as the Shitouhe Reservoir irrigation area, Baojixia irrigation area,

Fengjiashan Reservoir irrigation area, and Shibochuan Reservoir irrigation area, all of which increase the amount of groundwater extraction (Huang et al., 2017a).

### 5.1.3 Vegetation dynamics

     The DPRR results of NDVI (Figure 5) reveal a clear negative influence of vegetation dynamics on the PRR. Increased vegetation coverage enhances the interception capacity of vegetation canopies, resulting in more precipitation being stored and evaporated

within the vegetation canopy. This reduces the amount of water available for infiltration and surface runoff generation, thereby negatively affecting the precipitation-runoff dependencies. Furthermore, increased vegetation coverage leads to higher plant transpiration rates, where plants absorb soil moisture and release it into the atmosphere. As a result, there is a decrease in soil moisture available for baseflow generation, which contributes to the runoff. During precipitation events, drier soil retains more

precipitation, leading to a reduction in surface runoff generation. Additionally, increased vegetation strengthens the interception of median and low surface runoff in vegetated areas, resulting in a decrease in the amount of water contributing to runoff during the flow routing process (Buechel et al., 2022). Although vegetation interception of surface runoff increased infiltration to some extent, this increase is small compared to the reduction in runoff caused by increased vegetation. Moreover, vegetation interception of surface runoff also prolongs the flow routing time. With increased vegetation, the flow routing process is altered, and the same precipitation event may yield different runoff processes, thereby impacting the PRR (Wang et al., 2009; Chang et al., 2015).

The temperature in each sub-basin shows a significant upward trend (Figure 3), which is beneficial to vegetation growth. WR1, located in the upstream headwater area of the Wei River Basin, is characterized by its higher elevation and delicate ecological conditions, resulting in lower precipitation compared to the other sub-basins (Figure 3). Hence, vegetation has a more significant influence on intermediate and low time-scale surface runoff. In these regards, the impact of NDVI on the PRR is more significant in WR1 (Figure 5).

In the Wei River Basin, the scale of soil and water conservation projects has progressively expanded since the 1950s (Chang et al., 2015). These projects have been accompanied by the implementation of policies involving the conversion of farmland to forest and grassland (Feng et al., 2016; Zhang et al., 2008), resulting in a substantial increase in NDVI over time. The large-scale soil and water conservation projects in WR4 and WR5 lead to a greater increase in vegetation coverage compared to WR2 and WR3. However, the impact of these projects has diminished due to vegetation destruction associated with economic development since the 1990s (Wu et al., 2023), as evidenced by the results of the D-DPRR analysis (Figure 5c). Additionally, WR3 has a higher proportion of irrigated areas compared to WR2, and the planting patterns in these sub-basins typically involve winter wheat and summer corn. The planting patterns have an immediate effect on NDVI, thereby contributing to an important impact of NDVI on the PRR in WR3.

The impact of NDVI on the PRR varies depending on the time scale, particularly in WR4 and WR5 (Figure 5c). This variation arises from the differential effects of changes in vegetation coverage between urban and rural regions. In urban regions, vegetation is often introduced for urban greening, with plants transplanted from other regions. Once the planting is completed, the PRR exhibits a rapid response to changes in vegetation coverage. However, in rural regions, plant seedlings are typically planted, resulting in a slower impact on the PRR. The growth of planted vegetation in non-urban regions takes several years (depending on the vegetation type), and soil and water conservation projects are usually implemented in stages. Therefore, the influence of NDVI on the PRR in non-urban regions has higher uncertainty over time scales.

### 5.1.4 Anthropogenic influences

ISR, NTL, and POP are interconnected factors that represent the intensity of anthropogenic influences (Figure 4c). Impervious surface area is a crucial element of human settlements, directly indicating urbanization processes (Gong et al., 2020). With economic development, urban areas expand, infrastructure improves, production scales up, and the pursuit of a higher standard of living increases, leading to an increase in ISR and NTL (Ceola et al., 2019). While urban expansion generally accompanies population growth, in the case of small cities surrounding central cities, the opposite phenomenon may occur, especially in small cities surrounding central cities. Residents of small cities often migrate to central cities seeking a more convenient living environment or higher economic income. As a result, an increase in ISR and NTL, along with a decrease in population (POP), can be observed in the WR1 and WR4 (Figure 3). These differences have complex effects on regional PRR. In addition, Yang et al. (2024) investigated the impact of anthropogenic factors on water resources in China's nine major river basins, integrating data on

domestic, industrial, and irrigation water use. Wu et al. (2024) analyzed the effects of anthropogenic factors on water resources in the Yangtze River Basin, focusing on domestic, industrial, livestock, and irrigation activities. The findings of these studies indicate that population growth and urban expansion, along with behaviors such as local water extraction and inter-basin water transfers, significantly influence the PRR.

ISR exhibits a negative impact on PRR in all five sub-basins (Figures 5a and b). In WR4, the impact of ISR on PRR is second only to $BF$. This is because ISR alters the underlying surface conditions in the region, directly affecting multiple aspects of runoff generation in the hydrological cycle. Additionally, ISR exhibits similar levels of influence on PRR across different time scales, indicating consistent effects across varying time scales (Figure 5c). In the 1990s, the impact of ISR on PRR reached its highest level, possibly due to the development of infrastructure (Zhao et al., 2013).

In the five sub-basins, NTL exhibits similar levels of impact on PRR, except for WR4, where it shows a negative effect, while WR1, WR2, WR3, and WR5 show a positive effect (Figure 5a). This abnormal phenomenon may be attributed to the complex influence of NTL on PRR. NTL gradually increases with economic development and intensifies human activities, reflecting the intensity of nighttime human activities and energy consumption (Liao et al., 2017). Similar to ISR, NTL had a higher impact during the 1990s compared to other periods, as revealed by the D-DPRR results (Figure 5c).

The impact of POP on PRR primarily manifests through water consumption. WR1 and WR5 have a lower level of impact on PRR due to their smaller populations (Figures 3 and 5). However, WR5 has reservoirs such as Tuojiahe Reservoir and Zhengjiahe Reservoir (Figure 4d), where water in rivers is stored for agricultural irrigation and industrial and domestic use, resulting in reduced runoff. WR2 and WR4 exhibit higher levels of impact from POP on PRR (Figure 5). When water demand population aggregation exceeds the supply from rivers, groundwater is extracted to meet the demand, reducing baseflow generation and subsequently
decreasing runoff formation. Additionally, the growth in water consumption drives the construction of reservoirs, affecting the total volume of runoff. Constructing reservoirs increases regional infiltration, thereby increasing groundwater recharge, although this volume is small. WR3 has the largest population among the five sub-basins, but its impact level from population on PRR is low (Figure 3). Inter-catchment water transfer projects alleviate the increase in water demand caused by population growth, leading to a lower impact level of POP on PRR in WR3 (Zhang et al., 2011a).

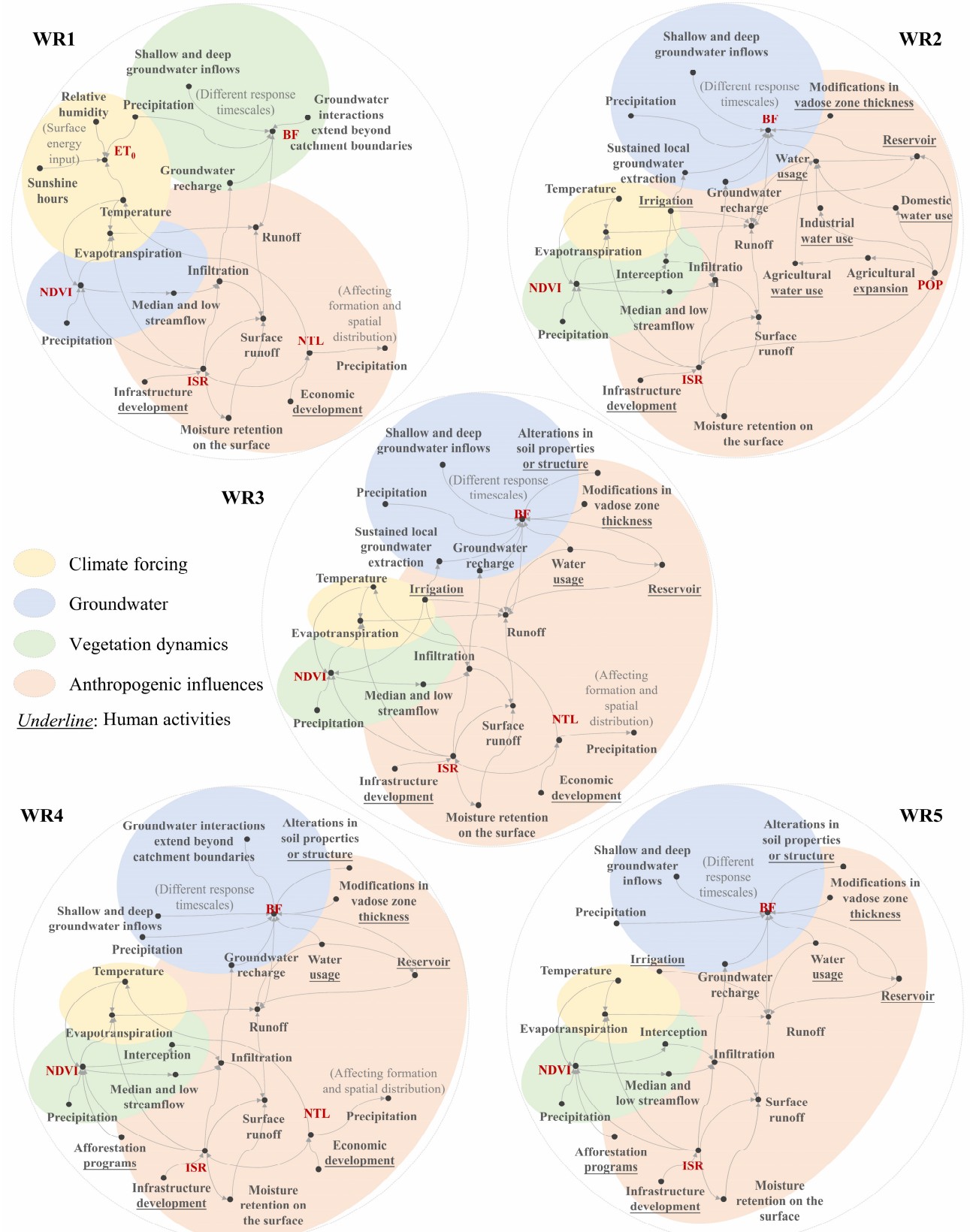

**Figure 6.** Causal loop diagram elucidating possible explanations of candidate driving factors in five sub-basins based on the driving levels and directions of influential factors on PRR.

### 5.1.5 Changes of potential processes controls in non-stationary conditions

Taking WR3 as an illustrative case (Figure 7), the potential processes controls for changes of precipitation-runoff relationships were investigated during two distinct sub-periods, denoted as sub-period 1 and sub-period 2, which correspond to pre- and post-hydrological regime shift in 2002. In accordance with the results of D-DPRR (Figure 5c), the impact of $ET_0$ on PRR is similar in sub-period 1 and sub-period 2. However, in sub-period 1, the impact of $ET_0$ on PRR exhibits a complex pattern, with alternating positive and negative effects. This complexity arises due to the potential processes controls associated with various meteorological elements (e.g., temperature, relative humidity, and wind) that influence $ET_0$. These meteorological elements are impacted by both global climate change and anthropogenic activities, such as the urban heat island effect, which affects local atmospheric circulation (Wai et al., 2017; Zhang et al., 2020a). As a result, the temporal effects of $ET_0$ on PRR become intricate and multifaceted. The impact of *BF* on PRR is the most significant compared to other factors. *BF* constitutes a crucial component of runoff, particularly during dry seasons in WR3 (Wang et al., 2009). During sub-period 1, *BF* exerts a high-level and positive impact on PRR. However, at the small-time scale of the sub-period 2, the impact of *BF* on the PRR becomes low-level and negative. This shift may be attributed to regional urbanization processes, leading to increased water consumption and large-scale groundwater extraction that alters *BF*'s influence on PRR (Huang et al., 2017a). NDVI consistently exerted a negative impact on the PRR in both sub-period 1 and sub-period 2. The increase in vegetation coverage led to heightened canopy interception, thereby reducing surface precipitation. However, the water storage capacity of the vegetation canopy is limited, and the transpiration process is constrained by the soil moisture content. Therefore, during sub-period 2, the impact of NDVI on precipitation-runoff relationships (PRR) exhibits a slight increase compared to sub-period 1. ISR and NTL serve as the indicators for assessing urbanization development and expansion (Gong et al., 2020). As such, their impact on the PRR is significant during sub-period 1, peaking around 1990 (see Figure 5c). This phenomenon can be attributed to the rapid growth of real estate and infrastructure in the WR3 during the 1990s (Zhao et al., 2013). However, due to data limitations, we were only able to evaluate the influence of POP on the PRR during sub-period 2, and its effect was found to be low.

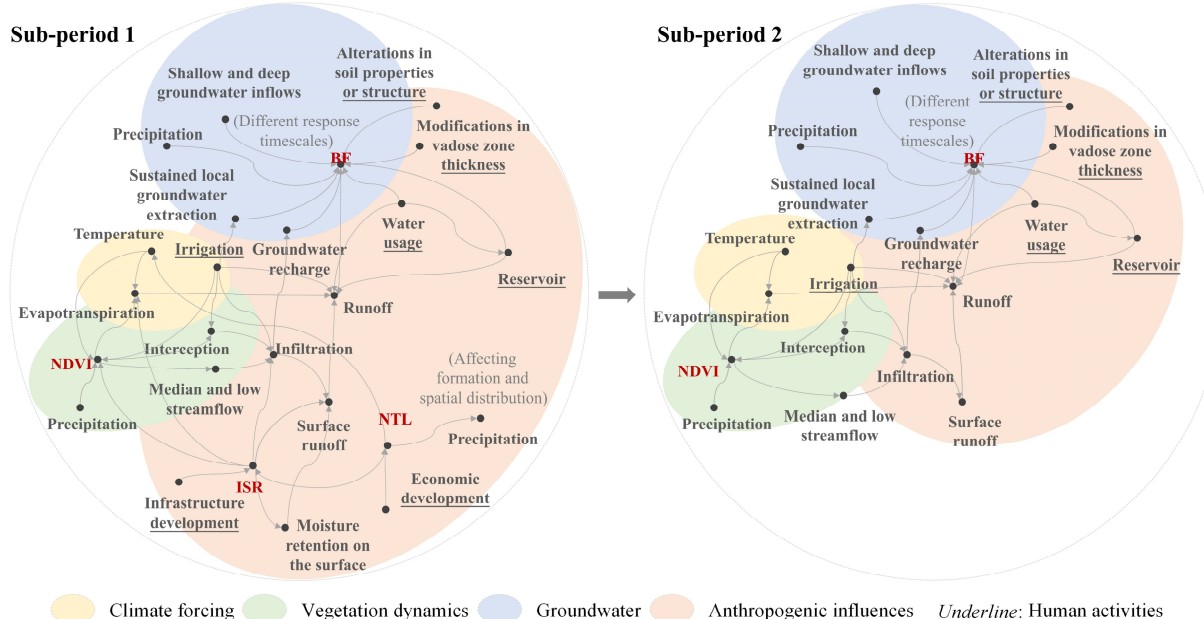

**Figure 7.** Causal loop diagram elucidating the changes of potential processes controls in WR3.

## 5.2 Comparison with previous literature

There are numerous studies on the impact of various factors on runoff changes in the Wei River basin. Gao et al. (2013) found that human activities contributed as much as 82.80% to the reduction in streamflow in the Wei River basin. Zhan et al. (2014a) used the SIMHYD model to partition the effects of climate change and human activities on surface runoff in the Wei River basin and found that the contribution of human activities to streamflow change was more than 65%. Zhan et al. (2014b) proposed the improved climate elasticity method to investigate the contributions of climate change and human activities to runoff changes in the Wei River basin, with results showing a climatic contribution to runoff decrease of 22–29% and a human contribution of 71–78%. Chang et al. (2015), using the VIC model, found that the percentages in runoff change due to climate change were 36%, 28%, 53%, and 10% in the 1970s, 1980s, 1990s, and 2000s, respectively. The percentages in runoff change caused by human activity were 64%, 72%, 47%, and 90%, respectively. It can thus be concluded that human activity has a greater impact on basin runoff than climate change factors. He et al. (2019), based on the Budyko framework, found that for the upper reaches of the Beiluo River, the contribution of land-use change variations to runoff reduction was 95.3%. Gao et al. (2020), using the SWAT model, found that in the Jing River basin, the influence of climatic factors decreased from 85.70 to 42.43%, while that of anthropogenic factors increased from 14.3% to 57.57% between 1961 and 2015. These studies indicate that human activities are the primary factor influencing PRR in the Wei River basin, which is consistent with the findings of this study. However, most existing research broadly categorizes influencing factors into climatic and anthropogenic factors, with some studies considering changes in potential evapotranspiration and land use as influencing factors. The quantitative assessment of human-induced impacts is often derived from the results of climatic factors without using specific data on human activities. In contrast, the method proposed in this study enables the exploration of the impact of individual factors on PRR.

## 5.3 Limitations and future research

Due to the intricate interplay of climate forcing, groundwater, vegetation dynamics, and anthropogenic influences on catchments, as well as the challenges associated with data collection on anthropogenic activities, it may not comprehensively account for all-inclusive or exhaustive driving factors in the changes of the PRR. In addition, the complex nonlinear interrelationships among the driving factors make it challenging to quantitatively synergistic nonlinear effects of various factors on PRR. In addition, the possible process explanations were verified by logical evidence and comparison with the findings in previous literature. In our subsequent investigation, we applied mutual information to validate the driving levels of the influencing factors on PRR. However, due to space limitations, a comprehensive elucidation of this index is elaborated in Supporting Information S7.

## 6 Conclusions

This study develops an integrated framework to explore the controls for changes in Precipitation-Runoff Relationships (PRR) in non-stationary environments. The introduction of the novel Driving index for changes in Precipitation-Runoff Relationships (DPRR) helps identify potential driving mechanisms influencing PRR within non-stationary hydrological processes. Taking the Wei River Basin as an example, by investigating candidate driving factors and incorporating computed driving levels and directions from DPRR and D-DPRR (dynamic) into a comprehensive conceptual model, possible explanations for PRR changes in this basin were derived as follows.

- Baseflow was identified as a predominant factor positively influencing PRR with significant uncertainty across various temporal scales. Potential evapotranspiration is a major driver of negative changes in PRR within sub-basins characterized by semi-arid climates and minimal human interference. Vegetation dynamics negatively impact PRR, with the extent of influence correlating directly with the scale of soil and water conservation efforts, displaying lower uncertainty.

- The impacts of urbanization, measured by Impervious Surface Ratio (ISR) and Night-Time Light (NTL), along with population density (POP), were found to vary, with ISR showing the most significant direct impact on PRR. These influences highlight the complex interplay between human activities and hydrological responses.

- The temporal analysis of DPRR values aligned well with historical shifts in hydrological regimes, suggesting that the proposed index can effectively capture the dynamics of PRR in response to non-stationary conditions. The possible process

explanations by the proposed index were supported by logical evidence and comparison with the findings in previous literature. Moreover, mutual information theory was applied to validate the main findings.

The proposed DPRR index provides a robust tool for comparing the influence of different drivers across varied catchment conditions and data-sequence lengths, offering a more nuanced understanding of hydrological responses. Thus, the integrated framework is able to provide a comprehensive understanding of hydrological processes, enabling informed decision-making for

sustainable water resource management in a basin. While this study is based on catchments in the Wei River Basin, it is expected that the issues discussed will be relevant to catchments in other parts of the world, particularly in areas with climate change and increasing anthropogenic pressures.

**Supplement link**

Supporting Information are provided.

**Author contributions**

Tian Lan devised the modelling concept. Tian Lan and Tongfang Li wrote the code, and prepared the original draft manuscript. Hongbo Zhang, Jiefeng Wu, Yongqin David Chen and Chong-Yu Xu provided supervision, and reviewed/edited the manuscript.

**Code availability**

All data set and model set up configurations have been reported in https://github.com/Suwine/DPRR.

**Competing interests**

The authors declare that they have no conflict of interest.

**Acknowledgments**

This study is financially supported by the National Natural Science Foundation of China (NSFC) (Grant No. 52209006, 52379003, and 52379013), the National Key R&D Program of China (2019YFC1510400), the China Postdoctoral Science Foundation (Grant

No. 2021M700018), the Fundamental Research Funds for the Central Universities, CHD (Grant No. 300102292104), and Research Council of Norway FRINATEK Project 274310.

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
