# Peer review of "Exploring the Potential Processes Controls for Changes of Precipitation-Runoff Relationships in Non-stationary Environments"

_Hydrology and Earth System Sciences, 2024_

## Author Response (AR1)

**Author's Response to anonymous referees for hess-2024-118**

"Exploring the Potential Processes Controls for Changes of Precipitation-Runoff Relationships in Non-stationary Environments" by Lan et al.

We would like to sincerely thank the anonymous reviewers for their time and thoughtful feedback on our manuscript. Their comments are highly encouraging and instrumental in improving the quality of our work. We have carefully considered and addressed all comments point-by-point, as detailed below. For clarity, the reviewers' comments are presented in black, and responses are provided in blue.

**Review 1**

**Comment on hess-2024-118, Anonymous Referee #1**

The topic "Exploring the Potential Processes Controls for Changes of Precipitation-Runoff Relationships in Non-stationary Environments" is valuable for hydrology. But this paper reads like a case study. The impacts of the study for the general hydrology and its novelty are not clear. The three main objectives of this study are developing an integrated framework, proposing a novel driving index for changes in DPRR, and establishing a holistic conceptual model. But developed the framework, driving index and conceptual model are also not clear and seem not innovative enough.

**Reply:** Thank you for affirming the value of this research in the field of hydrology. Regarding the **applicability** of this study, although we used the Wei River Basin, which is experiencing intensive anthropogenic activities and climate change as an example to demonstrate the proposed general framework, the general applicability of this study is emphasized in the *Discussion* and *Conclusion*. The specific revisions are in lines 405-409 and 591-594 of *Manuscript*.

Regarding the **innovation** of the main objectives of this study, the **first innovative point** is the proposal of a novel Driving Index for Changes in Precipitation-Runoff Relationships (DPRR) to quantify the driving levels and directions of factors influencing precipitation-runoff links. This index primarily addresses the limitations of traditional indices or models that

assume stationary conditions for assessing precipitation-runoff relationships in catchments exhibiting non-stationary behaviors. The **second innovative point** is the development of an integrated framework based on the proposed index, designed to explore the potential process controls on changes in precipitation-runoff relationships in non-stationary environments. The framework systematically includes detecting non-stationary processes, quantifying changes in PRR, assessing the driving levels and directions of potential influencing factors, analyzing hydrological responses to the temporal dynamics of driving factors, quantifying the nonlinear and intricate interplay among driving factors, and considering other anthropogenic influences such as large-scale surface water withdrawals from reservoirs and total water usage in the basin, including agricultural, industrial, and domestic sectors. The **third innovative point**, based on the aforementioned assessment results, is establishing a holistic conceptual model of catchment response to infer the potential processes controlling changes in precipitation-runoff relationships, which guides regional water use and resource allocation.

**Detail comments:**

1) To the best of my knowledge, the response of runoff to rainfall is non-linear, especially in the semi-arid regions, where infiltration excess runoff is dominant and the amount of runoff is sensitive to rainfall intensity. Rainfall as a major factor influencing the runoff coefficient should be considered, besides potential evapotranspiration.

**Reply:** We agree with the reviewer that the response of runoff to rainfall is non-linear and that precipitation is the most crucial factor in runoff generation. Given the importance of precipitation, we have used it as the input variable for our proposed Driving Index for Changes in Precipitation-Runoff Relationships (DPRR). Other factors are primarily used to explore their driving effects on the precipitation-runoff relationship.

2) In terms of anthropogenic activities, the constructions of check dams and reservoirs may be the more dominant factor influencing the runoff generation and the precipitation-runoff relationships in the region compared to ISR, NTL and POP.

**Reply:** We agree with the reviewer's viewpoint that the construction of check dams and reservoirs may be the dominant factor influencing precipitation-runoff relationships. This study

quantitatively investigated the impacts of reservoirs and various types of water usage (agricultural, industrial, and domestic) on precipitation-runoff relationships in *Section 4.5.2*. However, the collection of data for reservoirs and different types of water use in some catchments presented challenges, and some regions may not be influenced by reservoirs or dams. Additionally, acquiring long-term, continuous data on anthropogenic activities presents significant challenges. Remote sensing has proven to be an essential tool for identifying and assessing the temporal and spatial distributions of anthropogenic activities (An et al., 2024). Considering the study's applicability across various types of catchments, this study also uses various types of remote sensing data, including Impervious Surface Ratio (ISR) data, Night-Time Light (NTL) data, and Population (POP) data to comprehensively collect data on anthropogenic activities.

3) Vegetation dynamics are affected by both climate and afforestation, and how to distinguish them or consider their relationship with other factors?

**Reply:** We selected vegetation dynamics as a distinct control factor to explore its impact on PRR, primarily referencing the study by Fowler et al. (2022). In addition, the influence of climate change and human activities on vegetation dynamics, and consequently on the PRR, is highly complex. Therefore, we explore the impacts of climate forcing, anthropogenic influences, and vegetation dynamics on PRR, respectively.

4) Lines 96-97. What does the driving level and direction refer to?

**Reply:** Within a specified period, the driving level of DPRR signifies the influence level exerted by a particular factor on the correlation between precipitation and runoff during the period, and the driving direction of DPRR signifies whether a particular factor has positive or negative effects on the PRR during the period. The relevant content has been supplemented in lines 261-263 of *Manuscript*.

5) Figure 1-2. These sub-figures for each basin in Fig 1 and Fig 2(b) can be removed.

**Reply:** Thanks for the reviewer's comment. The sub-figures for each basin in Figure 1 have been removed to simplify the presentation of the study area. However, we have better explained

the content of Figure 2b, which is "Visual synthesis of selected process explanations for potential driving mechanisms of the changes in PRR under non-stationary processes depicting a general catchment affected by anthropogenic interference," and there are no "sub-figures for each basin".

6) Figure 3. It is inappropriate to put tables and graphs together in a figure.

**Reply:** Thanks for the reviewer's reminder. The table highlights the significant variation characteristics shown in Figure 3. The table and graphs have been separated.

7) 302-315. The heat map in Fig 4b is hard to understand and more detail is needed to explain. What's the relationship among these sub-figures. It seems inappropriate to put these in a figure.

**Reply:** Thanks for the reviewer's comment. The five sub-figures in Figure 4b correspond to the PRR results of the five basins. The content of the figures shows the PRR at various periods and time scales in each basin, that is, the DCCA values at various periods and time scales in each basin. The relevant content has been supplemented in the explanation (lines 326-327) of Figure 4b.

**References**

Fowler, K., Peel, M., Saft, M., Peterson, T. J., Western, A., Band, L., Petheram, C., Dharmadi, S., Tan, K. S., Zhang, L., Lane, P., Kiem, A., Marshall, L., Griebel, A., Medlyn, B. E., Ryu, D., Bonotto, G., Wasko, C., Ukkola, A., Stephens, C., Frost, A., Gardiya Weligamage, H., Saco, P., Zheng, H., Chiew, F., Daly, E., Walker, G., Vervoort, R. W., Hughes, J., Trotter, L., Neal, B., Cartwright, I., and Nathan, R.: Explaining changes in rainfall-runoff relationships during and after Australia's Millennium Drought: a community perspective, Hydrol. Earth Syst. Sci., 26, 6073-6120, 10.5194/hess-26-6073-2022, 2022.

**Review 2**

**Comment on hess-2024-118, Anonymous Referee #2**

1. Lines 75-82 the author wrote that hydrological models regionalize the PRR over a specific period, assuming minimal anthropogenic disturbance to simulate hydrologic processes. However, several hydrological models already consider anthropogenic effects on hydrological processes, such as WEP-L, watergap, and PCR-GLOBWB, which effectively incorporate anthropogenic impacts on hydrological processes into their simulations. So the statement "Hence, they may not be suitable for non-stationary hydrological processes" is inappropriate. Meanwhile, the list of PRR identification methods is not comprehensive. The author should highlight the similarities and differences between this study and these previous methods (hydrological model, machine learning, etc.), i.e., the research gaps, and emphasize the real advantages of the method in this study over the hydrological model, which has a physical mechanism, otherwise, it will be hard to be convincing.

**Reply:** We appreciate the reviewer's comment and agree with their suggestion. The statement "Hence, they may not be suitable for non-stationary hydrological processes" has been revised. Additionally, we have delineated the similarities and differences between this study and previous methods (hydrological model, machine learning, etc.), i.e., the research gaps, and emphasized the real advantages of the method in this study over the hydrological model, thereby addressing the identified research gaps.

**Compared to traditional data-driven and process-driven approaches, the primary advantages of the method (DPRR index) proposed in this study** are as follows: Continuous simulation of the PRR is accomplished through the application of data-driven hydrological models or process-based models with intricate model structures (Sikorska-Senoner and Quilty, 2021; Nayak et al., 2013). **Data-driven models** such as Artificial Neural Networks (ANN) and Linear Perturbation Model (LPM) rely on empirical analysis to produce corresponding outputs based on specific input data (Tanty and Desmukh, 2015; Nash and Barsi, 1983). **These models are heavily dependent on the characteristics of existing data and provide a relatively vague description of the hydrological cycle processes, failing to explicitly reflect the**

**impact of individual factors within these processes.**

On the other hand, process-based hydrological models such as TOPMODEL and SWAT can reveal more details and mechanisms of physical processes (Beven et al., 2021; Krysanova and White, 2015). However, **these models involve complex modeling processes and require careful selection based on adequate prior knowledge** (Reichl et al., 2009). Furthermore, **many traditional hydrological models regionalize the PRR over a specific period in stationary climatic and catchment conditions, assuming minimal human interference** (Yang et al., 2022; Westra et al., 2014). **As a result, they may not be suitable for non-stationary hydrological processes.**

With the advancement of hydrological models, the impact of human activities is increasingly incorporated into the simulation of hydrological processes. **Models such as WEP-L, WaterGAP, PCR-GLOBWB, and CWatM** not only simulate natural hydrological cycles but also effectively incorporate the influence of anthropogenic factors into their simulations (Jia et al., 2006; Müller Schmied et al., 2021; Sutanudjaja et al., 2018; Burek et al., 2020). However, **it is crucial to recognize that such hydrological models have high data requirements, limiting their application in regions lacking sufficient observational data** (Clark et al., 2016). For instance, **long-term, high-quality data on human activities are often difficult to obtain comprehensively**, and remote sensing datasets emerge as vital tools for the global analysis of human impacts on hydrological regimes. However, the **impact of human activities on streamflow represented by these data is complex and cannot be directly used as model input**.

Therefore, there is a need for a flexible and effective PRR technique that has lower data requirements while being able to explore the impact of individual or specific driving factors on PRR under non-stationary conditions. Compared with the above methods, **the DPRR index proposed in this study has the following advantages**: (1) the DPRR index has lower data requirements and offers a simple and flexible technique for identifying the potential impacts of driving factors on PRR. (2) The inputs are not strictly restricted by temporal and spatial constraints or by format, especially when assessing the impacts of anthropogenic activities on streamflow. (3) It also addresses the limitations of traditional process-driven hydrological models with the assumption of stationarity in the run, which assume stationary conditions

(Ammann et al., 2019; Jehanzaib et al., 2020). Hence, the proposed index offers crucial information for the hydrological cycle process driven by climate change or anthropogenic disturbances, which guides the construction of more robust hydrological models and the development of water resource management and allocation. (4) Considering the elimination of trends is a crucial step in accurately analyzing the relationships between complex non-stationary systems (Zhao et al., 2012), the DPRR removes the non-stationary effects by subtracting local trends with appropriate polynomial orders, ensuring the normality of input signals for cross-correlation analysis (Zebende, 2011). (5) The effect of external factors on PRR may lead to spurious cross-correlation estimations (Yuan et al., 2015). Hence, the developed index applies the theoretical foundations of the DPCCA technique to reveal the "intrinsic" relations between precipitation and runoff time series with potential influences of other factors removed, such as evapotranspiration, groundwater, land cover, and anthropogenic interference. (6) The DPRR index characterizes the potential driving mechanisms influencing PRR on different time scales, which can improve our understanding of hydrological responses to climate forcing and anthropogenic activities at various time scales. (7) The DPRR index provides the driving level and direction and allows for comparisons of the index values among different driving factors with inconsistent data-sequence lengths and across various types of catchments. The relevant content has been supplemented in lines 75-93 and 248-265 of *Manuscript*.

2. The author described the catchment response conceptual model with a very detailed relational network in Fig.2, Fig.6, and Fig.7. use a slightly more concise presentation? It might be more reader-friendly.

**Reply:** Thanks for the reviewer's comment. We have removed the symbols representing feedback types from Figures 2, 6, and 7 and eliminated some minor relationship lines to emphasize the primary relationships and structures. The detailed figures have been moved to lines 356-369 of *Supporting Information*.

3. How to validate the effectiveness of the constructed methods in the study for the

identification of non-stationary hydrological processes and their drivers? It seems unconvincing to describe its advantages over hydrological modeling only through text. There have been many studies analyzing the non-stationary hydrological processes in the Weihe River, and there is a need to compare with them to enhance the reliability of the results, as well as to quantify the uncertainties.

**Reply:** We agree with the comment of the Referee. Mutual information theory and techniques have been further applied to quantitatively validate the effectiveness of the constructed methods in the study. In addition, various studies on the impact of various factors on runoff changes in the Wei River basin are investigated and compared.

**The method based on mutual information theory for validating the proposed index in this study** is as follows. Entropy is a fundamental concept with wide-ranging applications across engineering and scientific disciplines (Mishra and Ayyub, 2019). It serves as a quantifiable metric for assessing signal uncertainty, simultaneously enabling the computation of mutual information between signal pairs. Mutual information (MI) is a measure of interdependence between variables (Cover, 1999). In these regards, we applied MI to develop an index for identifying the possible driving mechanisms in PRR using a nonlinear theory approach. By calculating mutual information between driving factors and precipitation (runoff), a Driving index for Precipitation-Runoff links with the nonlinear theory approach (DPRL) is developed and quantifies the nonlinear nature of their associations. Higher mutual information values signify stronger associations or interdependencies. The calculation procedure for the DPRL index is as follows.

**Step 1:** Involve three time series: the runoff time series denoted as $X_t$, the precipitation time series denoted as $Y_t$, and an influencing factor denoted as $Z_t$, where $t = 1, 2, \ldots, n$, and $n$ signifies the length of the time series. The initial computation entails deriving the cumulative frequency for each time series. Subsequently, the runoff time series is transformed into the following time series $Q_t$:

$$Q_t = \begin{cases} 1, & X_t \leq X^{20} \\ 2, & X^{20} < X_t \leq X^{40} \\ 3, & X^{40} < X_t \leq X^{60} \\ 4, & X^{60} < X_t \leq X^{80} \\ 5, & X_t > X^{80} \end{cases} \qquad (1)$$

where $X^{20}, X^{40}, X^{60}$, and $X^{80}$ correspond to $X_t$ when the cumulative frequencies are 20%, 40%, 60%, and 80%, respectively. Similar processing is applied to the precipitation time series $X_t$ and the influencing factor $Z_t$, resulting in the updated time series $W_t$ and $F_t$. These time series are discretized into five equidistant intervals to reduce the impact of noise while capturing a wider range of time series values across various magnitudes. Notably, the division into five equidistant boxes is a deduced outcome derived from rigorous comparative analyses and verifications (Franzen et al., 2020).

**Step 2:** Calculate the probability distribution functions for the time series:

$$\begin{cases} p(q_i) = \dfrac{\text{count}(q_i)}{n} \\ p(w_j) = \dfrac{\text{count}(w_j)}{n} \\ p(f_k) = \dfrac{\text{count}(f_k)}{n} \end{cases} \qquad (2)$$

where $p(q_i)$, $p(w_j)$ and $p(f_k)$ are the probability distribution functions of $Q_t$, $W_t$ and $F_t$ respectively; $\text{count}(q_i)$ , $\text{count}(w_j)$ and $\text{count}(f_k)$ represent the occurrences of numerical values in $Q_t$, $W_t$ and $F_t$, respectively; $i = 1,2, \ldots ,5$; $j = 1,2, \ldots ,5$; $k = 1,2, \ldots ,5$.

**Step 3:** The Shannon entropy of time series is calculated as follows:

$$H(Q_t) = - \sum_{i=1}^{5} p(q_i)\log_2 p(q_i) \qquad (3)$$

where $H(Q_t)$ is the Shannon entropy of $Q_t$. Here, entropy with a logarithm of base 2 is considered, such that entropy and related IT measures are in units of bits.

**Step 4:** Calculate the joint distribution functions as follows:

$$\begin{cases} p(q_i, f_k) = \dfrac{\text{count}(Q_t = q_i, F_t = f_k)}{n} \\ p(q_i, w_j) = \dfrac{\text{count}(Q_t = q_i, W_t = w_j)}{n} \end{cases} \qquad (4)$$

where $p(q_i, f_k)$ is the joint distribution function of $Q_t$ and $F_t$; $p(q_i, w_j)$ is the joint distribution function of $Q_t$ and $W_t$; $\text{count}(Q_t = q_i, F_t = f_k)$ is the number of simultaneous

occurrences of $Q_t = q_i$ and $F_t = f_k$ ; count$(Q_t = q_i, W_t = w_j)$ is the number of simultaneous occurrences of $Q_t = q_i$ and $W_t = w_j$.

**Step 5:** Given the influencing factor, the quantification of uncertainty within the sequence becomes feasible through the utilization of conditional entropy. This measure is computed as follows:

$$\begin{cases} H(Q_t|F_t) = \sum_{i=1}^{5} \sum_{k=1}^{5} p(q_i, f_k) \log_2 \dfrac{p(q_i, f_k)}{p(f_k)} \\ H(Q_t|W_t) = \sum_{i=1}^{5} \sum_{j=1}^{5} p(q_i, w_j) \log_2 \dfrac{p(q_i, w_j)}{p(w_j)} \end{cases} \tag{5}$$

where $H(Q_t|F_t)$ is the conditional entropy of $Q_t$ given $F_t$; $H(Q_t|W_t)$ is the conditional entropy of $Q_t$ given $W_t$.

**Step 6:** Mutual information $I(Q_t; F_t)$, quantifies the reduction in uncertainty of one variable when another variable is known. It is the difference between entropy and conditional entropy. The calculation for mutual information is as follows:

$$I(Q_t; F_t) = H(Q_t) - H(Q_t|F_t) = \sum p(q_t, f_t) \log_2 \frac{p(q_t, f_t)}{p(q_t)p(f_t)} \tag{6}$$

**Step 7:** The DPRL index is further updated as follows:

$$\text{DPRL}(t) = \frac{I(Q_t; F_t)}{H(Q_t|W_t) + 1} \tag{7}$$

where $I(Q_t; F_t)$ represents the mutual information between $Q_t$ and $F_t$. It quantifies the reduction in the uncertainty of $Q_t$ when $F_t$ is given, providing insights into their interdependence. With regard to the impact of precipitation on runoff, this index introduces the concept of conditional entropy $H(Q_t|W_t)$, accounting for the conditional uncertainty within runoff given precipitation. Furthermore, incorporating the notion of relative error, a modification is applied to the denominator by adding +1. This adjustment prevents the denominator from becoming exceedingly small, which may lead to anomalous metric values of the index.

[revised manuscript text omitted]

---

## Referee Report (RR1)

Reviewer Comments for revised Manuscript: " Exploring the Potential Processes Controls for Changes of Precipitation-Runoff Relationships in Non-stationary Environments "

To address the issue that present models assuming stationary conditions may result in incorrect streamflow forecasts, this study established a Driving index for changes in Precipitation-Runoff Relationships (DPRR). It provides possible process explanations for variations in precipitation-runoff relationships (PRR) using quantitative findings from inserting candidate driving factors into a holistic conceptual model. The study investigated the effects of climate forcing, groundwater, vegetation dynamics, and human activities on PRR in a non-stationary environment. This paper is important for hydrology since it provides a new perspective on hydrological processes and theoretical support for the building of long-term hydrological models. The manuscript is well-presented. Furthermore, the authors responded adequately to the prior two reviewers' suggestions, resulting in improvements to the paper's scientific value and study technique.

The study chooses ISR, NTL, and POP as factors to represent the impact of human activities on precipitation-runoff connections, and one factor each to investigate climate, groundwater, and vegetation dynamics. ISR, NTL, and POP are all intimately related in the context of urbanization. High ISR is typically associated with a greater population, higher levels of urbanization, and an increase in NTL. These three aspects interact; for example, population growth stimulates infrastructure expansion, which improves ISR and NTL. The writers are asked to explain why they chose these three elements to indicate human impact. Furthermore, ISR and NTL data are mostly acquired from remote sensing products, whereas POP data are primarily gathered through administrative planning, resulting in limited thorough observations of these data, particularly for distant historical periods. There are discrepancies between the timeframes of these data and other databases. The writers should explain the significance of these inconsistencies.

Within a given period, the driving level of DPRR represents the level of influence exerted by a specific factor on the correlation between precipitation and runoff during the period, while the driving direction of DPRR indicates whether a specific factor has positive or negative effects on the PRR during the period. Does this indicate that factors with a positive driving effect would increase runoff? Furthermore, because the results of DPRR and D-DPRR change over different durations, the influence of numerous factors on PRR remains unknown. In the version, vertical variations in the violin plot reflect the uncertainty in DPRR results. However, where is the uncertainty in D-DPRR data reflected?

Special comments
Line 133: It is recommended to convert the runoff volume of 8.09 billion m³ to runoff

depth in mm to facilitate comparison with the precipitation amount of 572 mm.

On the Impact of Human Activities on Hydrological Processes: The discussion section should incorporate insights from other models addressing similar topics to enhance the generalizability of the paper's conclusions. It is recommended to consult the following paper for a more comprehensive analysis:[1]Yang, X., Wu, F., Yuan, S., Ren, L., Sheffield, J., Fang, X., ... & Liu, Y. (2024). Quantifying the Impact of Human Activities on Hydrological Drought and Drought Propagation in China Using the PCR-GLOBWB v2. 0 Model. Water Resources Research, 60(1), e2023WR035443. [2] Wu, F., Yang, X., Cui, Z., Ren, L., Jiang, S., Liu, Y., & Yuan, S. (2024). The impact of human activities on blue-green water resources and quantification of water resource scarcity in the Yangtze River Basin. Science of the Total Environment, 909, 168550.

Lines 158-164: The selected datasets are in raster format, while time series data are used in the calculations. There is a lack of processing for the raster data.

Line 233: Verify the description of the value range here. Should this refer to the value range of bandwidth $\omega$?

---

## Author Response (AR2)

**Replies to Referee #2**

Dear Anonymous Referee #2,

Re: Manuscript entitled "Exploring the Potential Processes Controls for Changes of Precipitation-Runoff Relationships in Non-stationary Environments".

We greatly appreciate the Referee's comments. All suggestions are helpful to improve this manuscript. We have carefully studied, considered and responded to all comments point-by-point as follows. For clarity, all comments are given in black and responses are given in the blue text. All the comments and suggestions have been replied to below and will be addressed in the revision.

Yours sincerely,

Tongfang Li
E-mail: tongfangli@chd.edu.cn

To address the issue that present models assuming stationary conditions may result in incorrect streamflow forecasts, this study established a Driving index for changes in Precipitation-Runoff Relationships (DPRR). It provides possible process explanations for variations in precipitation-runoff relationships (PRR) using quantitative findings from inserting candidate driving factors into a holistic conceptual model. The study investigated the effects of climate forcing, groundwater, vegetation dynamics, and human activities on PRR in a non-stationary environment. This paper is important for hydrology since it provides a new perspective on hydrological processes and theoretical support for the building of long-term hydrological models. The manuscript is well presented. Furthermore, the authors responded adequately to the prior two reviewers' suggestions, resulting in improvements to the paper's scientific value and study technique.

The study chooses ISR, NTL, and POP as factors to represent the impact of human activities on precipitation-runoff connections, and one factor each to investigate climate, groundwater, and vegetation dynamics. ISR, NTL, and POP are all intimately related in the context of urbanization. High ISR is typically associated with a greater population, higher levels of urbanization, and an increase in NTL. These three aspects interact; for example, population growth stimulates infrastructure expansion, which improves ISR and NTL. The writers are asked to explain why they chose these three elements to indicate human impact. Furthermore, ISR and NTL data are mostly acquired from remote sensing products, whereas POP data are primarily gathered through administrative planning, resulting in limited thorough observations of these data, particularly for distant historical periods. There are discrepancies between the timeframes of these data and other databases. The writers should explain the significance of these inconsistencies.

**Reply:** We greatly appreciate the positive evaluation for this study. The influence of anthropogenic factors on the Precipitation-Runoff Relationship (PRR) is complex and multifaceted. Although ISR, NTL, and POP are closely related, their impacts on PRR vary significantly. Impervious surfaces, composed of man-made structures that impede natural water infiltration, are a key component of urban residential areas (Gong et al., 2020). The high ISR in urbanized regions affects surface energy and water balance, influencing extreme precipitation and flooding events (Lu et al., 2019). NTL products offer comprehensive insights into the impact of human presence and economic development on water resources (Ceola et al., 2019). Higher POP typically leads to increased urbanization, resulting in more impervious surfaces such as buildings and roads, along with heightened water demand (Fang and Jawitz, 2019). These three factors were selected to comprehensively explore the influence of anthropogenic factors on PRR.

Compared to traditional hydrological models, the Driving index for changes in Precipitation-Runoff Relationships (DPRR) requires less data and provides a straightforward and effective technique to identify potential driving factors affecting PRR. The proposed DPRR index quantifies driving levels and enables comparative analysis between different driving factors with varying data lengths and across diverse types of basins. In addition to ISR, NTL, and POP, this study further investigates anthropogenic influences over the study period, particularly reservoirs and their associated large-scale irrigation zones, to better understand the driving mechanisms of anthropogenic factors.

*The relevant content is presented in Section S5 of the Supporting Information as well as Sections 2.2 and 3.2 of the manuscript.*

Within a given period, the driving level of DPRR represents the level of influence exerted by a specific factor on the correlation between precipitation and runoff during the period, while the driving direction of DPRR indicates whether a specific factor has positive or negative effects on the PRR during the period. Does this indicate that factors with a positive driving effect would increase runoff? Furthermore, because the results of DPRR and D-DPRR change over different durations, the influence of numerous factors on PRR remains unknown. In the version, vertical variations in the violin plot reflect the uncertainty in DPRR results. However, where is the uncertainty in D-DPRR data reflected?

**Reply:** Thank you for the Referee's reminding. During a given period, the driving direction of the DPRR indicates the influence of a specific factor on PRR. However, it is essential to note that DPRR is constructed based on DCCA and DPCCA and represents a statistical approach. The DPRR index captures only the driving level and direction of factors influencing PRR changes and does not directly reflect specific hydrological processes or water fluxes within the hydrological cycle. Therefore, a factor with a positive driving influence does not imply an increase in runoff.

DPRR elucidates potential driving mechanisms affecting PRR at various timescales, enhancing our understanding of hydrological responses to climatic forcing and human activities across different temporal scales. The dispersion of the DPRR index along the vertical axis of violin plots reflects its uncertainty across timescales. At a specific timescale, the D-DPRR index provides clarity on the influence of factors on PRR at distinct time points. The results of the D-DPRR index are presented using heatmaps, where the vertical axis represents different timescales and the horizontal axis represents specific time points. Heatmaps prominently illustrate the variations in the D-DPRR index across timescales, highlighting the uncertainty of different factors' impacts on PRR.

Special comments

Line 133: It is recommended to convert the runoff volume of 8.09 billion m³ to runoff depth in mm to facilitate comparison with the precipitation amount of 572 mm.

**Reply:** Thank you for the Referee's suggestion. We change "8.09 billion m³" to "60 mm". The relevant descriptions will also be revised in Section 2.1 of the revised manuscript.

On the Impact of Human Activities on Hydrological Processes: The discussion section should incorporate insights from other models addressing similar topics to enhance the generalizability of the paper's conclusions. It is recommended to consult the following paper for a more comprehensive analysis:[1]Yang, X., Wu, F., Yuan, S., Ren, L., Sheffield, J., Fang, X., ... & Liu, Y. (2024). Quantifying the Impact of Human Activities on Hydrological Drought and Drought Propagation in China Using the PCR-GLOBWB v2. 0 Model. Water Resources Research, 60(1), e2023WR035443. [2] Wu, F., Yang, X., Cui, Z., Ren, L., Jiang, S., Liu, Y., & Yuan, S. (2024). The impact of human activities on blue-green water resources and quantification of water resource scarcity in the Yangtze River Basin. Science of the Total Environment, 909, 168550.

**Reply:** We agree with the Referee's suggestion. Yang et al. (2024) investigated the impact of anthropogenic factors on water resources in China's nine major river basins, integrating data on domestic, industrial, and irrigation water use. Wu et al. (2024) analyzed the effects of anthropogenic factors on water resources in the Yangtze River Basin, focusing on domestic, industrial, livestock, and irrigation activities. The findings of these studies indicate that population growth and urban expansion, along with behaviors such as local water extraction and inter-basin water transfers, significantly influence the PRR. Relevant studies have been referenced and discussed in the *Discussion* section.

Lines 158-164: The selected datasets are in raster format, while time series data are used in the calculations. There is a lack of processing for the raster data.

**Reply:** Thank you for the Referee's suggestion. The processing of NDVI data involves averaging the raster data within each study area. For ISR data, the number of impervious surface rasters within each study area is divided by the total number of rasters. NTL data is processed by summing the nighttime light intensity within each study area. Similarly, POP data is processed by summing the population within each study area. The relevant content has been supplemented in Section 2.2 of the revised manuscript.

Line 233: Verify the description of the value range here. Should this refer to the value range of bandwidth $\omega$?

**Reply:** Thank you for the Referee's reminding. The search space for $w$ is set to [0.05, 1.95] with a step size of 0.05. The relevant content has been revised in Section 3.2 of the manuscript.

References

Ceola, S., Laio, F., and Montanari, A.: Global-scale human pressure evolution imprints on sustainability of river systems, Hydrol. Earth Syst. Sci., 23, 3933-3944, https://doi.org/10.5194/hess-23-3933-2019, 2019.

Fang, Y. and Jawitz, J. W.: The evolution of human population distance to water in the USA from 1790 to 2010, Nat. Commun., 10, 430, https://doi.org/10.1038/s41467-019-08366-z, 2019.

Gong, P., Li, X., Wang, J., Bai, Y., Chen, B., Hu, T., Liu, X., Xu, B., Yang, J., Zhang, W., and Zhou, Y.: Annual maps of global artificial impervious area (GAIA) between 1985 and 2018, Remote Sens. Environ., 236, 111510, https://doi.org/10.1016/j.rse.2019.111510, 2020.

Lu, M., Xu, Y., Shan, N., Wang, Q., Yuan, J., and Wang, J.: Effect of urbanisation on extreme precipitation based on nonstationary models in the Yangtze River Delta metropolitan region, Sci. Total Environ., 673, 64-73, https://doi.org/10.1016/j.scitotenv.2019.03.413, 2019.

Wu, F., Yang, X., Cui, Z., Ren, L., Jiang, S., Liu, Y., and Yuan, S.: The impact of human activities on blue-green water resources and quantification of water resource scarcity in the Yangtze River Basin, Sci. Total Environ., 909, 168550, https://doi.org/10.1016/j.scitotenv.2023.168550, 2024.

Yang, X., Wu, F., Yuan, S., Ren, L., Sheffield, J., Fang, X., Jiang, S., and Liu, Y.: Quantifying the Impact of Human Activities on Hydrological Drought and Drought Propagation in China Using the PCR-GLOBWB v2.0 Model, Water Resour. Res., 60, e2023WR035443, https://doi.org/10.1029/2023WR035443, 2024.

---

## Author Response (AR3)

**Replies to Editor**

Dear Editor,

Re: Manuscript entitled "Exploring the Potential Processes Controlling Changes in Precipitation-Runoff Relationships in Non-stationary Environments".

We sincerely appreciate the editor's review and suggestions regarding the manuscript. We have carefully considered and responded to all recommendations point-by-point as follows. For clarity, all recommendations are given in black and responses are given in blue text. All the recommendations have been replied to below and revisions have been made in the manuscript.

Yours sincerely,

Tongfang Li
E-mail: tongfangli@chd.edu.cn

Recommendation:

All comments have been addressed, and the paper is ready for publication following a thorough proofreading. For instance, the phrase 'Potential Processes Controls of' in the title (and throughout the paper) could be revised to 'potential processes controlling changes in' to improve readability.

**Reply:** We greatly appreciate the recommendation. We have made revisions to the relevant content in the manuscript, as detailed in Table 1.

**Table 1.** The lines and content revised in the manuscript.

| Number of lines | Before revision | After revision |
|---|---|---|
| 1 | Potential Processes Controls for Changes of | Potential Processes Controlling Changes in |
| 114 | potential processes controls for changes in | potential processes controlling changes in |
| 117 | potential processes controls on changes in | potential processes controlling changes in |
| 528 | Changes of potential processes controls in | Potential processes controlling changes in |
| 529 | potential processes controls for changes of | potential processes controlling changes in |
| 551 | changes of potential processes controls in | potential processes controlling changes in |